# Chiron-o1: Igniting Multimodal Large Language Models towards Generalizable Medical Reasoning via Mentor-Intern Collaborative Search

**Haoran Sun[1,2*], Yankai Jiang[1* ✉], Wenjie Lou[1], Yujie Zhang[2]**
**Wenjie Li[1,3], Lilong Wang[1], Mianxin Liu[1], Lei Liu[2], Xiaosong Wang[1✉]**
[1]Shanghai Artificial Intelligence Laboratory
[2]Fudan University
[3]Shanghai Jiao Tong University
`jiangyankai@pjlab.org.cn, wangxiaosong@pjlab.org.cn`

## Abstract

Multimodal large language models (MLLMs) have begun to demonstrate robust reasoning capabilities on general tasks, yet their application in the medical domain remains in its early stages. Constructing chain-of-thought (CoT) training data is essential for bolstering the reasoning abilities of medical MLLMs. However, existing approaches exhibit a deficiency in offering a comprehensive framework for searching and evaluating effective reasoning paths towards critical diagnosis. To address this challenge, we propose Mentor-Intern Collaborative Search (MICS), a novel reasoning-path searching scheme to generate rigorous and effective medical CoT data. MICS first leverages mentor models to initialize the reasoning, one step at a time, then prompts each intern model to continue the thinking along those initiated paths, and finally selects the optimal reasoning path according to the overall reasoning performance of multiple intern models. The reasoning performance is determined by an MICS-Score, which assesses the quality of generated reasoning paths. Eventually, we construct MMRP, a multi-task medical reasoning dataset with ranked difficulty, and Chiron-o1, a new medical MLLM devised via a curriculum learning strategy, with robust visual question-answering and generalizable reasoning capabilities. Extensive experiments demonstrate that Chiron-o1, trained on our CoT dataset constructed using MICS, achieves state-of-the-art performance across a list of medical visual question answering and reasoning benchmarks. Codes are available at https://github.com/Yankai96/Chiron-o1

## 1 Introduction

Multimodal Large Language Models (MLLMs) [8, 23, 26, 34, 49, 72] have demonstrated prominent performance in a wide range of tasks, including image captioning, visual question answering, and video analysis. Recent breakthroughs in MLLMs have also shed new light on the development of general-purpose medical AI. Remarkable efforts have been made to adapt existing MLLMs to address complex clinical tasks through supervised fine-tuning (SFT) on carefully curated multimodal medical instruction fine-tuning datasets [7, 28, 30, 65, 69]. However, most current medical MLLMs rely on a direct-prediction paradigm, which produces brief, immediate answers to problems and often overlooks the interleaved image–text reasoning processes essential for real-world clinical scenarios. This critical shortcoming, namely the inability to perform deep multimodal reasoning analysis of

---

[*]Equal contribution.
[✉] Corresponding authors.

medical data, has given rise to an urgent need for the development of more sophisticated medical MLLMs capable of tackling complex clinical challenges and offering improved diagnostic support.

Building on recent advancements in reinforcement learning (RL) optimization, several promising approaches [24, 39, 47] have leveraged Group Relative Policy Optimization (GRPO) [43] to elicit reasoning capabilities in models, introducing multiple R1-series medical MLLMs. Nevertheless, these models continue to exhibit limited performance in answering real-world clinical questions. As pointed out by recent studies [75], although RL training improves performance by biasing the model's output distribution toward reward-yielding trajectories, it fails to generate novel reasoning paradigms. Consequently, the performance ceiling of the R1 models has long been constrained by their underlying base MLLM. One potential solution is to incorporate high-quality chain-of-thought (CoT) annotations into SFT to bolster the model's reasoning capabilities and foster novel reasoning paradigms. However, unlike general-domain tasks in which large-scale CoT datasets can be crowdsourced, medical reasoning demands domain-specific logical structures and clinical expertise. Curating such medical CoT datasets is prohibitively expensive and time-consuming. Furthermore, the absence of standardized evaluation metrics to ensure the validity of generated CoT processes also remains a critical barrier to step-by-step visual reasoning annotation. Therefore, it is crucial to develop an effective and sophisticated method that can generate intermediate reasoning steps toward the final answer and evaluate the step-by-step visual reasoning quality.

Motivated by the aforementioned challenges, we propose Mentor-Intern Collaborative Search (MICS), a new multi-model collaborative searching strategy designed to generate effective step-by-step CoT data. The core idea of MICS is leveraging multiple knowledgeable mentor models to collaboratively search for reasoning paths, while evaluating the searched paths based on feedback from intern models. The entire search process aligns with the logic of how a mentor guides interns, where the effectiveness of the mentor's guidance is validated by whether interns can correctly solve problems based on the provided instructions, i.e., initialized reasoning paths. We integrate the valuable insights of multiple mentor models, retaining effective steps and discarding low-quality or hallucinated ones to identify the correct reasoning paths. Along the searching, we propose an MICS-score to enable the evaluation of CoT data quality, to further facilitate the efficient construction of multimodal reasoning data. Building on the proposed methods, we develop MMRP, a multimodal medical reasoning dataset comprising three subsets: simple question–answer (QA) pairs, image–text alignment annotations, and MICS-generated multimodal CoT data for complex clinical scenarios. Thus, we employ a novel curriculum learning paradigm that progressively infuses medical knowledge, from fundamental concepts to complex cases in MMRP, thereby enhancing the model's reasoning capabilities. Finally, we achieve Chiron-o1, a general-purpose multimodal medical model with multimodal CoT reasoning capabilities through a stage-wise SFT with the composed curriculum.

We conduct a comprehensive evaluation on seven benchmarks, containing both in-domain and out-of-domain scenarios, to rigorously assess the performance of Chiron-o1. The results indicate that Chiron-o1 exhibits robust multimodal reasoning capabilities, outperforming the SOTA medical reasoning models across all benchmarks. Our contributions can be summarized as follows:

- We present **MICS**, a multi-model collaborative search strategy that facilitates the generation of effective step-by-step CoT data.

- We construct **MMRP**, a high-quality multimodal medical dataset comprising QA pairs, image-text alignment data, and effective reasoning paths for complex medical VQA problems, spanning 12 imaging modalities and 20 body systems.

- We develop **Chiron-o1**, a new multimodal medical model that demonstrates outstanding reasoning abilities in handling both in-domain and out-of-domain complex clinical problems.

- We demonstrate that our approach achieves competitive performance compared to previous SOTA medical MLLMs through extensive experiments across multiple benchmarks.

## 2 Related Works

**Reasoning in Medical MLLMs.** As multimodal reasoning demand grows, frameworks and techniques have evolved, extending the reasoning capabilities of Large Language Models (LLMs) [18, 51, 52] to tackle more complex general-purpose vision-language tasks [54, 72, 9, 3]. Techniques such as prompt tuning [76], SFT [44], and RL [11] have emerged as critical methods for boosting the

reasoning capabilities of MLLMs. Given its potential to enhance model's performance on complex tasks, reasoning ability has also garnered significant attention in medical domains. Models like HuatuoGPT-o1 [7] and Baichuan-M1 [55] are dedicated to enhancing the ability to solve medical problems through rigorous and transparent reasoning paths. However, equipping MLLMs with reasoning capabilities by integrating multimodal medical information remains an open and challenging problem. Recent studies [35, 64] have explored designing well-crafted prompts to simulate the reasoning process from medical problems to correct answers, often involving multiple doctor roles or functional modules. However, the applicability of the aforementioned methods is limited. Meanwhile, as DeepSeek-R1 significantly enhances model reasoning capabilities through GRPO [18], an increasing number of studies follow this paradigm to adapt RL to the medical domain. Med-R1 [24], utilizing the GRPO strategy, demonstrates superior performance across different image modalities, enhancing the generalizability of MLLMs in medical reasoning. Similarly, MedVLM-R1 [39] adopts an RL framework to encourage the discovery of human-interpretable reasoning paths. However, these RL-based models rely solely on fixed reward functions (e.g., format and accuracy), neglecting the evaluation of reasoning paths, which can easily lead to superficial or hallucinated reasoning processes. In contrast, our proposed method leverages a collaborative search to identify high-quality reasoning paths with minimal hallucinations, thereby enhancing reasoning abilities.

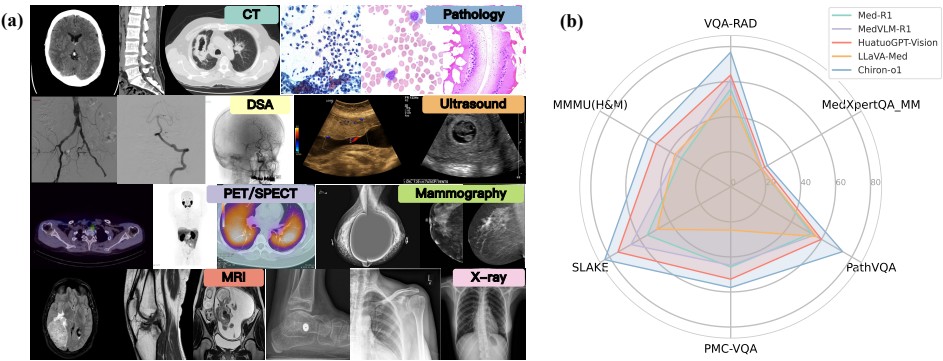

Figure 1: **Overview of the MMRP Dataset and Chiron-o1 Performance.** (a) The MMRP dataset encompasses 12 imaging modalities and 20 body systems. (b) Chiron-o1 achieves SOTA performance across various benchmarks compared to existing multimodal medical models.

**Construction of CoT Datasets.** The development of CoT datasets for SFT is crucial in multimodal reasoning, setting it apart from traditional models [1, 29, 82, 33] by emphasizing the reasoning process rather than merely providing final answers [61, 63]. Research in this field can be categorized into prompt-based, plan-based, and learning-based methods [62], each contributing uniquely to the CoT generation in multimodal contexts. Prompt-based methods utilize carefully designed prompts to guide models in generating CoT during inference. Simple prompts, such as "think step-by-step" [21], initiate creation of reasoning paths for multimodal tasks, while advanced approaches establish a clear reasoning workflow by specifying detailed objectives and procedures for each step [10, 15, 37]. Few-shot prompts often include reasoning examples [6, 80], offering flexibility in resource-limited scenarios or when rapid responses are needed. Plan-based methods enable dynamic exploration of reasoning paths, enhancing adaptability. For example, MM-ToT [17] combines GPT-4 [23] and Stable Diffusion [42], using depth-first and breadth-first searches to select optimal solutions. HoT [70] encapsulates interconnected thoughts within hyperedge structures, while AGoT [68] builds reasoning graphs integrating visual data. Additionally, learning-based methods embed CoT construction within model training or fine-tuning. PCoT [56] optimizes this approach for CoT generation, whereas MC-CoT [48] improves reasoning capabilities in smaller-scale models via majority voting during training. However, as constructing medical CoT requires specific logic and domain expertise, the aforementioned methods cannot be effectively transferred to the medical field. In other words, the construction of multimodal CoT in the medical domain remains an underexplored stage. Existing studies [64] primarily rely on manual annotations or model-generated reasoning paths without any evaluation. Our proposed method takes a step further by addressing these challenges by leveraging an automatic and collaborative search strategy to identify high-quality reasoning paths.

# 3 Methods

In this section, we elaborate on our method for eliciting effective reasoning capabilities in MLLMs to tackle complex clinical problems by constructing multimodal medical CoT datasets and applying SFT. Specifically, we propose a novel curriculum learning scheme tailored for training medical MLLMs. It involves leveraging the constructed dataset MMRP to progressively inject medical knowledge into MLLMs, advancing from foundational principles to complex problems. Concretely, we develop QA and image-text alignment datasets to strengthen the model's foundational capabilities in the medical domain. Subsequently, we introduce the MICS strategy to generate high-quality CoT data for complex medical scenarios. MICS also incorporates a novel evaluation metric designed to assess the effectiveness of reasoning paths. Ultimately, using the aforementioned datasets, we perform stage-wise SFT to train our model, Chiron-o1, which demonstrates robust reasoning performance.

## 3.1 MMRP Dataset

Currently, numerous websites and publications provide complex clinical cases for educational purposes, such as Radiopaedia [16], BMJ Case Reports [13], and World Journal of Clinical Cases [19]. These platforms aim to enhance learners' knowledge and, more crucially, translate it into tangible improvements in individual competence for daily clinical practice. Utilizing the data released in [81] (mainly from Radiopaedia [16]), we additionally aim to recompose training data for robust reasoning capabilities. Specifically, we construct the MMRP dataset with three subsets. Part 1 of the MMRP is designed to create a QA dataset at the level of clinical trainees. Initially, we collect over 60K+ cases containing pure text information, encompassing neurological disorders, cardiovascular abnormalities, skeletal system diseases, and more. Subsequently, QA pairs are synthesized in a segmented manner based on the information richness of the cases. Furthermore, we compose Part 2 of the MMRP with aligned image-text pairs. Unlike other methods that generate synthetic image captions [7], texts in this subset are entirely sourced from authentic medical imaging analyses in the educational site, covering various imaging findings such as masses and lesions, anatomical abnormalities, inflammatory changes, and more. Further details about Parts 1 and 2 of MMRP are provided in the Appendix A.1 and A.2. Next, we construct multimodal CoT data in Part 3 of the MMRP using MICS to enhance the reasoning capabilities of MLLMs.

## 3.2 Mentor-Intern Collaborative Search for Effective Reasoning

Besides the data in Parts 1 and 2, the MMRP dataset also aims to construct a subset of multimodal reasoning paths by leveraging valuable clinical case information. To explore effective reasoning paths, we propose the MICS, a multi-model collaborative reasoning path search strategy. The strategy emulates the mentor-intern dynamic, where effective guidance depends on interns' performance in correctly solving problems using the prompts. The core idea is to leverage powerful mentor models to iteratively search for and identify valuable reasoning steps, shown in Figure 2. The effectiveness of these steps is determined by feedback from intern models. Inspired by CoT [53, 67] and existing process reward models [57, 59, 79], we believe that models can address problems by thinking in a step-by-step manner, thereby deriving answers in a reasonable and understandable way. To evaluate the quality of each step, we define its value as the potential of leading to the correct answer.

We denote the mentor models as $\{\theta_1, \ldots, \theta_n\}$ and the intern models as $\{\beta_1, \ldots, \beta_m\}$. The former are role-played with generalist models, while the latter typically is composed of open-sourced models, often much smaller in size. During searching, at the step $k$, an intermediate reasoning step generated by a $\theta_n$ is represented as $s_{k,n}$ accordingly. The searching begins at the <start point> and subsequently explores effective reasoning paths through iterative search. It involves three key operations.

**1) Collaborative search for reasoning paths by multiple mentors** The objective of this operation is to combine the knowledge acquired by multiple mentor models to collaboratively search for effective reasoning paths. This approach not only enhances the diversity of reasoning paths but also mitigates the risk of a single model unilaterally solving a problem, particularly when the model's understanding of the problem is biased. Specifically, starting from the <start point> or the optimal path selected in operation 3), each mentor model $\theta$ generates a complete solution, thereby deriving the correct answer. In practice, this process involves $\theta$ acting as a "completer", extending the reasoning path prefix. The entire process can be formalized as follows:

$$\{s_{1,x}, s_{2,y}, ..., s_{k,z}\} \sim \text{MICS}_k(\cdot \mid \theta_1, \ldots, \theta_n, \beta_1, \ldots, \beta_m) \tag{1}$$

$$s_{k+1,v} \sim \theta_v(\cdot \mid P, Q, A, \{s_{1,x}, s_{2,y}, ..., s_{k,z}\}, I) \tag{2}$$

where $\text{MICS}_k$ denotes the search process up to the step $k$, and $\{s_{1,x}, s_{2,y}, ..., s_{k,z}\}$ represents the all reasoning steps searched thus far, serving as the reasoning path prefix for the subsequent search. $P$, $Q$, $A$, and $I$ represent patient information, questions, ground truth, and corresponding images.

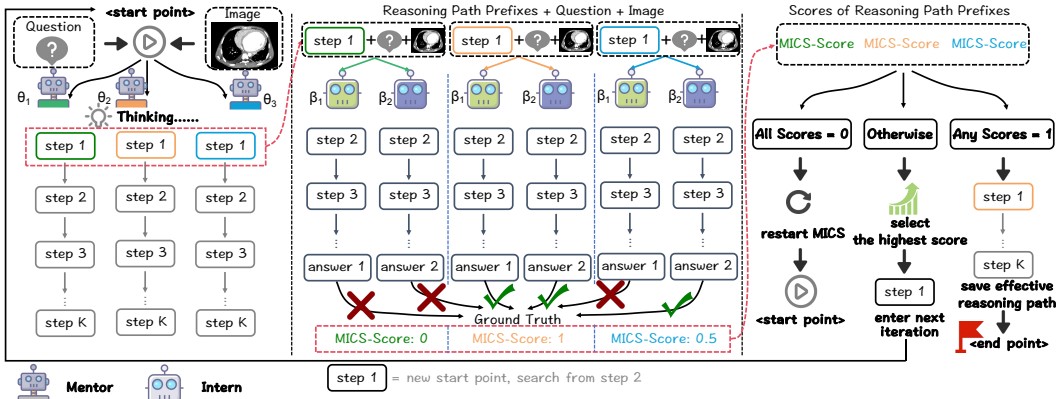

Figure 2: **Framework of the MICS Strategy.** MICS enables search for effective reasoning paths through collaboration between mentor and intern models until the maximum search depth is reached or early-stopping conditions are met. $\theta$ denotes the mentor model, and $\beta$ denotes the intern model. The example of CoT construction using MICS is provided in the Figure 7.

**2) Evaluation of reasoning paths based on MICS-Score** We argue that relying on closed-source MLLMs to evaluate the steps in a reasoning path is arbitrary [71]. Instead, the value of a reasoning path explored by a mentor model $\theta$ should be assessed by intern models $\beta$. Fundamentally, this operation involves the intern models $\beta$ completing the entire reasoning process based on the prompt (reasoning path prefix) provided by the mentor model $\theta$. The effectiveness of the reasoning path prefix is then measured by comparing the answer decoded by the intern model $\beta$ against the ground truth. This validation logic aligns with the principle that "a mentor's guidance is effective only if interns can correctly solve problems based on the provided prompts. Specifically, we employ lightweight models from diverse families with varying temperatures as intern models $\beta$ to improve sampling diversity and randomness. Ultimately, the quality of a reasoning path is defined as the frequency with which it achieves the correct answer. The entire evaluation process is formalized as follows:

$$\tilde{s}_{>k+1,w}, \tilde{a}_w \sim \beta_w(\cdot \mid Q, \{s_{1,x}, s_{2,y}, ..., s_{k,z}, s_{k+1,v}\}, I), \quad w \text{ ranging from } 1 \text{ to } m \tag{3}$$

$$\text{MICS-Score}(s_{\leq k+1}) = \frac{\text{number of LLM}(\tilde{a}_w, A)}{\text{number of } \tilde{a}_w} \tag{4}$$

where $\tilde{s}_{>k+1,w}$ and $\tilde{a}_w$ denote the reasoning path completed by the intern model $\beta_w$ and the final answer, respectively. Subsequently, we employ an LLM [31] to compare the generated answer with the ground truth to determine its correctness. The proportion of interns answering correctly is calculated as the value score (MICS-Score) of the reasoning path prefix.

**3) Selection of the optimal reasoning path** Following the previous operations, we obtain the reasoning prefixes generated by different mentor models in the current search iteration, along with their corresponding value scores. In each search iteration, we select the reasoning path with the highest score as the reasoning prefix for the next search, continuing until the maximum search depth is reached. MICS strategy is not solely guided by high-value outcomes but also incorporates exploratory attributes [5]. If multiple reasoning path prefixes in a iteration yield the same scores, i.e., $score(\{s_{1,x}, s_{2,y}, ..., s_{k,z}, s_{k+1,v}\})$ equals $score(\{s_{1,x}, s_{2,y}, ..., s_{k,z}, s_{k+1,r}\})$. We prioritize the reasoning step generated by a mentor model $\theta_r$ that was not selected in previous iterations as the current step. Additionally, during the search process, we implement two "early stopping" mechanisms. First, if all mentor models in a search iteration reach a score of zero, the data is marked as a search failure, triggering a subsequent re-search. Second, if the current reasoning path prefix enables all intern models to derive the correct answer, the search is terminated. Particularly, if multiple full-score

reasoning paths exist, we calculate the competitiveness of the corresponding mentor models, as follows:

$$\text{Competitiveness}(\theta_v, n) = \prod_{i=1}^{n} \text{MICS-Score}(\{s_{<i}, s_{i,v}\}) \tag{5}$$

where $s_{<i}$ denotes the reasoning path searched up to the i-th iteration, and Competitiveness$(\theta_v, n)$ represents the competitiveness score of the mentor model at the (n+1)-th search, calculated by multiplying the value scores obtained in previous search iterations. A higher competitiveness score indicates that the mentor model possesses clearer insights into the problem compared to other models.

In summary, the MICS strategy leverages three key operations to iteratively search until the maximum depth is reached or a full-score reasoning path is identified. By applying MICS to complex clinical problems, we can construct a set of high-quality, step-by-step reasoning triplets, comprising questions, reasoning paths, and answers, thereby enabling MLLMs to learn to reason.

### 3.3 Chiron-o1: Multi-Stage SFT for Reasoning Emergence

Inspired by curriculum learning [4, 45, 60], we design a three-stage model training strategy based on different task types. This strategy emulates the human learning process, progressing from simple to complex knowledge acquisition. Specifically, we begin by training the model on pure text medical QA tasks, gradually enabling it to answer simple medical questions and provide concise explanations. Subsequently, the model is trained to become familiar with the characteristics of medical images, achieving effective image-text alignment through authentic clinical images and their analyses. Unlike methods that generate image findings from images and captions [7], we utilize the original image analyses from cases as the supervision signal during this stage to minimize the introduction of fabricated information. Building on these stages, the final training stage leverages high-quality CoT data, generated by the MICS strategy in complex medical scenarios, to stimulate the model's reasoning capabilities. However, the MMRP dataset alone is insufficient to endow the model with comprehensive multimodal medical knowledge, which could hinder the emergence of reasoning abilities. Therefore, we incorporated several commonly-used Visual Question Answering (VQA) datasets [7, 20, 25, 32, 78] to bolster the model's foundational capabilities. In practice, all curated datasets are mixed in specific proportions during the training process to train the model effectively.

$$D_l = f(\text{VQA\_Set}, \sum_{i=1}^{l} \text{MMRP\_Part\_}l), \quad l \text{ ranging from 1 to 3} \tag{6}$$

$$L(\Omega, l) = \sum_{(Q,Y,A) \in D_l} \log \Omega(Y, A \mid Q) \tag{7}$$

where $f(\cdot)$ denotes the fusion of the utilized datasets in specific proportions, as detailed in the Appendix C. We iterate through all SFT data (consisting of triplets of questions, reasoning paths, and answers, like $(Q, Y, A)$) from the predefined dataset $D_l$ to train the MLLM $\Omega$.

## 4 Experiments and Results

### 4.1 Experiment Settings

**Benchmarks** To evaluate our model, we utilize two types of medical benchmarks: VQA benchmarks and reasoning benchmarks. (1) The former focuses on testing the model's foundational capabilities, such as visual understanding. We utilized the VQA-RAD [25], SLAKE [32], PathVQA [20], PMC-VQA [78], and the Health & Medicine track subset of the MMMU dataset [74]. (2) The latter is designed to evaluate the model's reasoning capabilities when addressing complex problems. This benchmark includes MedXpertQA_MM (multimodal subset) [83] and the MMRP test set. MedXpertQA_MM is divided into two subsets emphasizing reasoning and understanding, respectively. The MMRP test set encompasses two pre-divided subsets of pure text and multimodal reasoning data.

**Implementation Details** In our experiments, the MICS strategy employs three mentor models: ChatGPT-4o [23], Gemini 2.5 Pro Preview [50], and Qwen2.5-VL-72B-Instruct [3]. To validate the searched reasoning paths, we designate three open-source models as intern models, each queried

under two temperatures (0.3 and 1.2): Qwen2-VL-7B [58], Qwen2.5-VL-7B [3], and InternVL3-8B [9]. We employ DeepSeek-V3 [31] as the "judge" to compare intern answers against the ground truth. To balance search efficiency and reliability, the maximum search depth is set to 4.

For model training, we adopt InternVL3-8B as the base model and apply LoRA fine-tuning [22] with AdamW as the optimizer, a learning rate of 4e-5 with cosine decay, a batch size of 16, and bfloat16 mixed precision. Training is conducted on eight NVIDIA A100 GPUs with DeepSpeed ZeRO-1 [41]. The three curriculum stages require approximately 12 hours, 12 hours, and 48 hours, respectively, totaling 72 GPU-hours. The training corpus consists of the MMRP dataset and several commonly used medical VQA datasets, with detailed composition provided in Appendix C.

**Baseline Methods & Evaluation Metrics**   We compare Chiron-o1 with the following three types of baseline models: (1) General MLLMs, including high-performance general vision models, both closed-source (GPT series [23], Gemini series [50]) and open-source (LLaVA series [33], Qwen series [3]). (2) Medical MLLMs, comprising vision models pretrained on specific medical corpora, such as LLaVA-Med [28], GMAI-VL [30], and HuatuoGPT-Vision [7]. (3) Medical Reasoning MLLMs, including Med-R1 [24], MedVLM-R1 [39], and ChestX-Reasoner [14], which are fine-tuned using GRPO [43]. For evaluation metrics, we use choice accuracy to measure model performance on VQA benchmarks. For the open-ended questions requiring reasoning in Part 3 of the MMRP dataset, BERT-Score [77] is employed to assess the semantic similarity between the model's final answer and the ground truth, while MICS-Score is used to evaluate the effectiveness of the reasoning path.

Table 1: **Main Results on Medical VQA Benchmarks.** Our model achieves SOTA performance across various benchmarks. **Bold** denotes the highest score, and underline denotes the second-highest.

| Methods | VQA-RAD | SLAKE | PathVQA | PMC-VQA | MMMU(H&M) | AVG |
|---|---|---|---|---|---|---|
| *Close-Source SOTA* | | | | | | |
| **Gemini-1.5-Pro** [49] | 60.3 | 72.6 | 70.3 | 52.3 | 47.9 | 60.7 |
| **Gemini-2.5-Pro** [12] | 71.3 | 80.5 | 73.9 | **61.1** | **57.1** | 68.8 |
| **GPT-4o-mini** [23] | 55.8 | 50.4 | 48.7 | 39.6 | – | 48.6 |
| **GTP-4o** [23] | 54.2 | 50.1 | 59.2 | 40.8 | – | 52.1 |
| *Open-Source SOTA* | | | | | | |
| **LLaVA-v1.5-7B** [33] | 54.2 | 59.4 | 54.1 | 36.4 | 38.2 | 48.5 |
| **LLaVA-v1.6-13B** [33] | 55.8 | 58.9 | 51.9 | 36.6 | 39.3 | 48.5 |
| **LLaVA-v1.6-34B** [33] | 58.6 | 67.3 | 59.1 | 44.4 | 48.8 | 55.6 |
| **Yi-VL-34B** [73] | 53.0 | 58.9 | 47.3 | 39.5 | 41.5 | 48.1 |
| **Qwen-VL-Chat** [2] | 47.0 | 56.0 | 55.1 | 36.6 | 32.7 | 45.5 |
| *Medical MLLM* | | | | | | |
| **LLaVA-Med** [28] | 51.4 | 48.6 | 56.8 | 24.7 | 36.9 | 43.7 |
| **Med-Flamingo** [38] | 45.4 | 43.5 | 54.7 | 23.3 | 28.3 | 39.1 |
| **RadFM** [66] | 50.6 | 34.6 | 38.7 | 25.9 | 27.0 | 35.4 |
| **GMAI-VL** [30] | 66.3 | 72.9 | – | 54.3 | 51.3 | 61.2 |
| **HuatuoGPT-Vision-7B** [7] | 63.8 | 74.5 | 59.9 | 52.7 | 49.1 | 60.0 |
| **HuatuoGPT-Vision-34B** [7] | 68.1 | 76.9 | 63.5 | 58.2 | 54.4 | 64.2 |
| *Reasoning Medical Model* | | | | | | |
| **Med-R1** | 55.9 | 55.1 | 53.3 | 45.8 | 32.7 | 48.6 |
| **MedVLM-R1** | 61.4 | 65.9 | 55.2 | 44.8 | 35.5 | 52.6 |
| **ChestX-Reasoner** | 70.9 | 70.0 | 66.7 | 38.5 | 49.5 | 59.1 |
| **InternVL3-2B** | 68.3 | 65.9 | 65.2 | 49.1 | 38.4 | 57.4 |
| **Chiron-o1-2B** | 75.4 | **85.3** | 70.3 | 54.3 | 42.1 | $65.5^{+8.1}$ |
| **InternVL3-8B** | 73.1 | 71.1 | 67.9 | 53.2 | 52.1 | 63.5 |
| **Chiron-o1-8B** | **76.8** | 83.2 | **74.0** | 57.5 | 54.6 | $\mathbf{69.2}^{+5.7}$ |

## 4.2   Main Results

**Performance on the Medical VQA Benchmarks**   Medical models should strive to further enhance their performance on VQA tasks while improving reasoning capabilities. Therefore, to evaluate the foundational performance of Chiron-o1, we first test it on several commonly-used VQA benchmarks. The results, compared against other mainstream models, are summarized in Table 1. Compared to the baseline models, Chiron-o1 achieves significant performance improvements across five benchmarks.

Table 2: **Results on Medical Reasoning Benchmarks.** ACC represents accuracy, and MICS-Score refers to the evaluation metric described in Equation 4. ∗ indicates pure text reasoning models.

| Model | MMRP (Pure Text) | MedXpertQA_MM (Reasoning) | MedXpertQA_MM (Understanding) | MMRP (Reasoning) | | |
|---|---|---|---|---|---|---|
| | ACC | ACC | ACC | ACC | Bert-Score | MICS-Score |
| MedReason* | 79.2 | – | – | – | – | – |
| HuatuoGPT-o1* | 85.1 | – | – | – | – | – |
| Med-R1 | 72.7 | 20.1 | 20.8 | 28.1 | 83.4 | 22.5 |
| MedVLM-R1 | 77.5 | 21.7 | 20.0 | 31.2 | 83.5 | 23.5 |
| **Chiron-o1-2B** | 90.6 | 19.8 | 23.1 | 43.8 | 88.2 | 32.2 |
| **Chiron-o1-8B** | **92.1** | **23.3** | **25.1** | **58.4** | **90.4** | **49.4** |

Specifically, Chiron-o1-8B and Chiron-o1-2B outperform their respective baseline models by an average of 5.7% and 8.1%. This demonstrates that our reasoning-focused model further enhances visual understanding and question-answering capabilities. Next, we compared Chiron-o1 with SOTA models that were neither pretrained nor fine-tuned on multimodal medical datasets. The results in Table 1 indicate that our model easily surpasses these large-scale MLLMs. Subsequently, Chiron-o1 outperforms most medical MLLMs across all benchmarks. It even achieves comparable performance to HuatuoGPT-Vision-34B, which has four times the parameters, on the PMC-VQA and MMMU (H&M) benchmarks, while significantly surpassing it on the remaining benchmarks, with an overall average improvement of 5%. Finally, we observe that existing medical reasoning models, such as Med-R1 and MedVLM-R1, tend to lose basic VQA capabilities while focusing on reasoning, as shown in Table 1. In contrast, Chiron-o1-8B outperforms them on VQA benchmarks by an average of 20.6% and 16.6%, respectively. Overall, Chiron-o1 demonstrates competitive performance across benchmarks, highlighting its versatility in medical image understanding and question answering.

**Performance on Medical Reasoning Benchmarks** We further evaluate the performance of Chiron-o1 on reasoning benchmarks, including in-domain (MMRP) and out-of-domain (MedXpertQA_MM) datasets. [40] posits that training multimodal reasoning models may compromise their textual reasoning capabilities. Therefore, we first assess Chiron-o1 on the pure text reasoning data of the MMRP dataset. The results in Table 2 demonstrate that our model significantly outperforms the visual reasoning models Med-R1 and MedVLM-R1. Even compared to pure text reasoning models, Chiron-o1 surpasses MedReason and HuatuoGPT-o1 by 12.9% and 7%, respectively. These results strongly validate that our proposed MICS and stage-wise training strategy effectively enhance the model's text reasoning capabilities. In the multimodal reasoning domain, Chiron-o1 excels in both the accuracy of final answers and the effectiveness of reasoning paths compared to other reasoning models. The results in Table 2 indicate that Chiron-o1-8B consistently outperforms MedVLM-R1 (the best among other reasoning models) by 27.2% in accuracy and 6.9% in semantic similarity. The effectiveness of reasoning paths is another key focus. Compared to all reasoning models, Chiron-o1-8B achieves the highest score of 49.4% on the MICS-Score. Our model not only excels on in-domain datasets but also demonstrates robust performance on out-of-domain benchmarks. Chiron-o1-8B outperforms Med-R1 and MedVLM-R1 on MedXpertQA_MM by an average of 3.75% and 3.35%, respectively. Figure 3 illustrates that Chiron-o1 can engage in deep and reasonable reasoning for real-world complex clinical problems, ultimately providing correct answers.

### 4.3 Ablation Studies

**Effect of MICS Strategy** We apply the MICS strategy to build the MMRP multi-task dataset to improve the model's multimodal reasoning. An ablation study on Chiron-o1-8B evaluates its impact. Unlike MICS, the vanilla method enables the mentor model to generate reasoning directly, bypassing the evaluation of intern models. Table 3 shows that excluding reasoning data (first row) or using reasoning paths constructed by vanilla method(second and third rows) degrades performance. Compared to Chiron-o1-8B, the former reduces average performance by 3% across four benchmarks, and the latter by 2.7%. Furthermore, Figure 4 illustrates the differences in effectiveness between reasoning paths constructed using MICS and the vanilla method across three distinct medical scenarios. We compute the MICS-Score step-by-step for an entire reasoning path to evaluate the value of reasoning prefixes and analyze the trend of MICS-Score changes (categorized into four types, with a monotonically increasing score indicating an effective trend and the others deemed low-value,

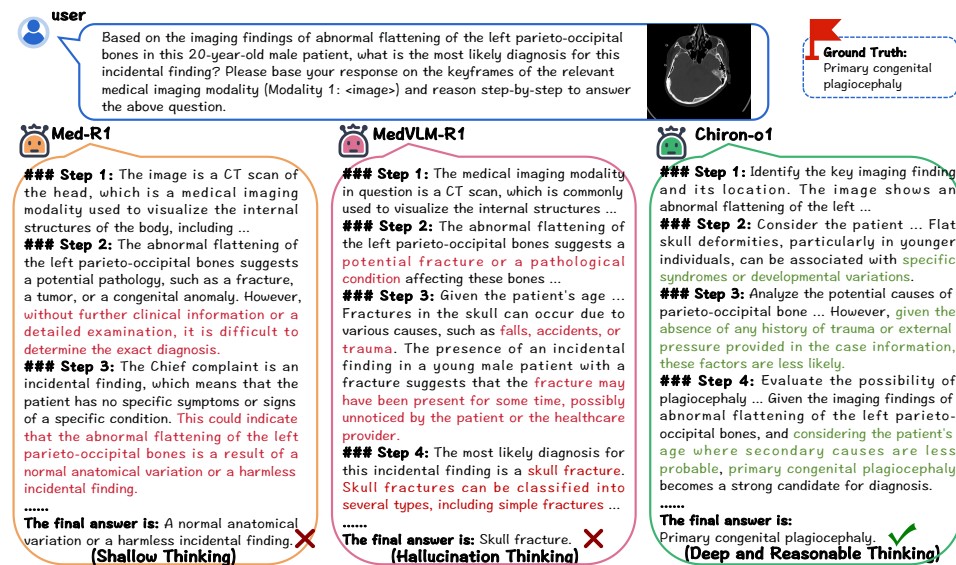

Figure 3: **Case Study on the MMRP Test Set.** Compared to other multimodal medical reasoning models, Chiron-o1-8B demonstrates the ability to generate deep and reasonable reasoning paths, leading to correct answers. Due to page limitations, details are provided in the Appendix G.

details provided in the Appendix B). As depicted in Figure 4, the proportion of effective reasoning paths identified by MICS significantly surpasses that of the vanilla method. These results underscore the critical role of the MICS strategy in searching for effective reasoning paths.

Table 3: Ablation Studies on Training Set. We examine how different dataset combinations affect performance. MMRP(▲) indicates direct use of mentor models for reasoning path search, while MMRP(✓) denotes MICS-based search. VQA shows whether medical VQA datasets are included in training. QC indicates whether quality control is applied to MICS-searched reasoning paths.

| MMRP | VQA | QC | VQA-RAD | SLAKE | MMMU(H&M) | MedXpertQA_MM |
|---|---|---|---|---|---|---|
|  | ✓ | — | 73.6 | 80.3 | 49.7 | 23.2 |
| ▲ |  | — | 71.3 | 76.9 | 52.3 | 23.4 |
| ▲ | ✓ | — | 75.7 | 81.2 | 51.6 | 23.6 |
| ✓ |  | ✓ | 72.4 | 75.0 | 51.1 | 24.0 |
| ✓ | ✓ |  | 74.3 | 82.6 | 49.3 | 22.6 |
| ✓ | ✓ | ✓ | 76.8 | 83.2 | 54.6 | 24.2 |

**Effect of Training Set Settings** During training, we incorporated VQA datasets into the training set to bolster the model's foundational visual understanding capabilities. To evaluate their contribution to model performance, we compared a model trained without VQA datasets to Chiron-o1-8B. The results in Table 3 (second and fourth rows) indicate that VQA data significantly enhances the model's performance on simpler benchmarks. Specifically, performance on VQA-RAD and SLAKE increased by an average of 5% and 7.3%, respectively. However, this operation has limited impact on complex and reasoning benchmarks. We further investigated whether quality control of the MMRP reasoning data could improve the model's reasoning performance. As shown in Table 3 (fifth row), quality control enhances the model's accuracy on MMMU (Health & Medicine) and MedXpertQA_MM, improving from 49.3% to 54.6% (+5.3%) and from 22.6% to 24.2% (+1.6%), respectively. These results effectively validate that our specific setting of the training set is both effective and reasonable.

**Effect of Training Strategy** In Section 3.3, we proposed a stage-wise training strategy for fine-tuning the model. To evaluate its impact, we trained models using only subsets of the stages and compared their performance on several benchmarks against Chiron-o1-8B. First, for the complex VQA benchmark (MMMU Health & Medicine), Figure 5(a) shows that models trained with Stage 1, Stage 1+2, or Stage 3 alone exhibit significant performance gaps compared to Chiron-o1-8B, with reductions of 6.9%, 6.2%, and 3.5%, respectively. Next, we observed that omitting the third stage

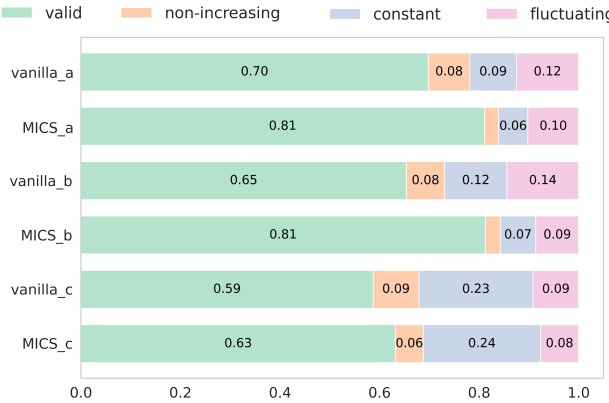

Figure 4: **Ablation Studies on MICS.** Contribution of the MICS strategy to reasoning path score trends, with a, b, and c denoting three clinical scenarios (Appendix A.3). "vanilla" refers to directly generating reasoning paths using the mentor model without evaluation.

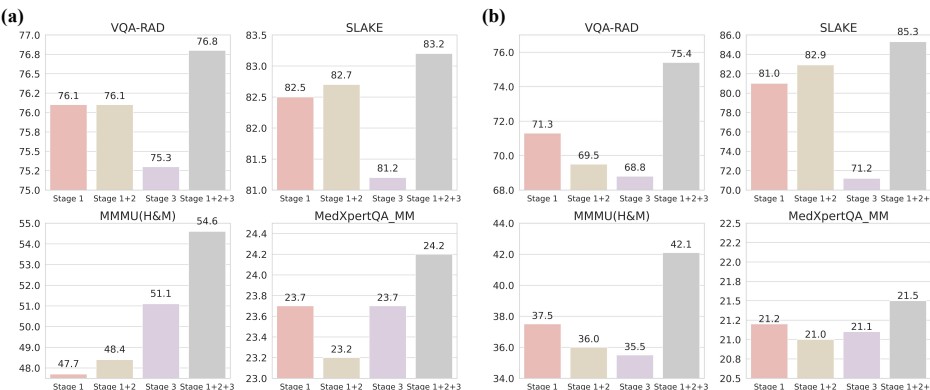

Figure 5: **Ablation studies on the Model Training Strategy.** Figure (a) and (b) present results for Chiron-o1-8B and Chiron-o1-2B, respectively. The comparison highlights the advantage of the proposed stage-wise curriculum over alternative training schemes.

leads to a more pronounced degradation in reasoning capabilities. Specifically, models trained with Stage 1 or Stage 1+2 perform, on average, 3.1% and 0.25% lower than those trained with Stage 3 alone on MMMU (Health & Medicine) and MedXpertQA_MM. Similarly, training solely with Stage 3 (without reinforcing visual question-answering capabilities) severely impairs performance on VQA-RAD and SLAKE compared to other ablation models.

## 5 Conclusion

This paper introduces MICS, a multi-model collaborative search strategy that generates high-quality multimodal medical reasoning data by preserving valid reasoning steps and eliminating incorrect ones, enhancing medical CoT construction efficiency. With a stage-wise fine-tuning approach, we use MICS to create MMRP, a multi-task medical reasoning dataset with varying difficulty. This enables Chiron-o1, a robust multimodal model, to achieve SOTA performance across benchmarks. We believe this work advances medical CoT data construction and improves reasoning in medical MLLMs.

## Acknowledgments

This work is funded by the National Key R&D Program of China under grants No.2022ZD0160700 and is supported by Shanghai Artificial Intelligence Laboratory.

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

# A    Deatails of MMRP

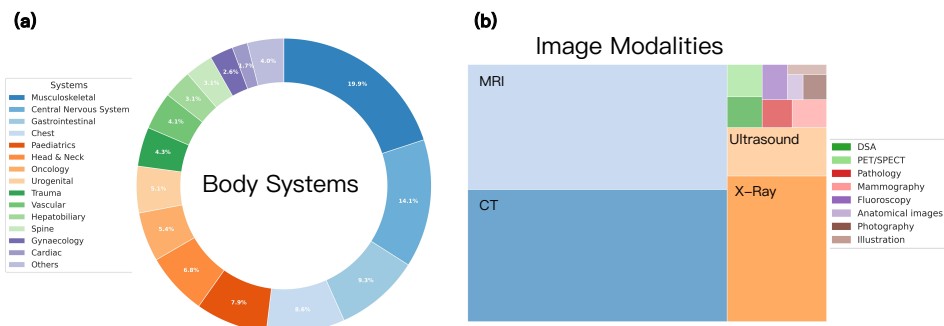

Figure 6: Distribution of MMRP across various systems and modalities.

## A.1    Part 1: Starting with Simple Knowledge Injection!

The "simple" refers not only to task difficulty but also to the number of modalities involved. It is widely acknowledged that a model's robust reasoning capabilities are built upon extensive pretraining with vast datasets. Accordingly, Part 1 of the MMRP dataset is designed to create a text-only QA dataset at the level of medical interns or clinical interns. Initially, we collected 60,789 cases containing textual information. Due to missing critical information in the medical imaging analysis $C$ or case summary and discussion $D$, incomplete cases were filtered out. Subsequently, each case was categorized into four levels based on the token count of $C$ and $D$, as calculated by a tokenizer [3], to assess the information richness of the case. Cases with greater information content were deemed suitable for generating more QA pairs, while those with limited information were used sparingly to avoid redundancy in QA pairs. We employed DeepSeek-V3 [31] as the LLM to generate QA pairs in a segmented manner, including the question $q$, question options $o$, the correct answer $a$, and the rationale $r$, as outlined below:

$$token\_number = \text{tokenizer}(C, D) \tag{8}$$

$$(q, o, a, r) \sim \text{LLM}( \bullet \, | \, prompt_x(P, C, D) ) \, , prompt_x \text{ depends on } token\_number \tag{9}$$

where $P$ refers to basic information of the patients. The $prompt_x$ is provided in the Figure 8. Among the generated QA pairs, some were of low quality, irrelevant to medicine, or failed to parse correctly. These inappropriate QA pairs were removed. Additionally, we filtered out data that could cause ambiguity or hallucinations (e.g., content containing phrases like "this case" or "this image") to ensure that the generated questions could be answered independently. Ultimately, Part 1 of the MMRP dataset comprises 57,630 QA quadruplets.

## A.2    Part 2: Understanding the Content Conveyed by Images!

To enable the model to perform reasoning in multimodal medical scenarios, it is essential to align multimodal information using image-text pairs derived from real medical images and their corresponding analyses. Unlike Part 1, this subset of data includes only annotated cases (i.e., cases with explicitly identified key information, such as lesion locations, on specific slices), totaling approximately 3K cases and covering 12 distinct imaging modalities. As the imaging analysis $C$ may reference multiple key findings associated with the same image, we employed file-level MD5 hashing to deduplicate images [46] and mitigate hallucination risks. Additionally, low-resolution images were filtered out, ensuring a minimum dataset resolution of 196×196. During training, we treat each modality's image-text pairs within a case as a single data unit. However, if a modality contains an excessive number of keyframe images, it may lead to misalignment between images and text [27] or cause out-of-memory issues during training and inference. Consequently, we excluded image-text pairs with more than 10 keyframes.

In constructing the training data, we designed two distinct alignment rules: "coarse alignment" and "precise alignment." The former emphasizes holistic understanding of multiple images, ensuring that

the sequence of image descriptions corresponds to the order of the images. The latter focuses on mapping specific keywords in the text to their corresponding images (e.g., like "... heart failure (image x) ..."). Notably, the image description information in this subset is directly sourced from authentic medical imaging analyses in Radiopaedia [16], rather than being synthetically generated. Given that even SOTA MLLMs struggle to accurately interpret medical images, Part 2 of the MMRP dataset is designed to minimize hallucination risks. Consequently, we constructed a dataset of 5,878 triplets, each comprising a keyframe image, a question about the image, and the imaging analysis, to serve as image-text alignment data.

### A.3   Part 3: Learning to Reason for Complex Problems via MICS!

**Data Collection**   Unlike Parts 1 and 2 described earlier, this subset aims to synthesize multimodal reasoning processes by leveraging valuable clinical case information. Specifically, during CoT synthesis, we utilize a broader range of authentic clinical information to minimize unfounded model hallucinations, as opposed to relying solely on QA pairs. For the collected cases, we designed three typical clinical QA scenarios using DeepSeek-V3 [31], resulting in a total of 8,328 complex visual QA pairs, as outlined below (see Figure 10, 11 and 12 for prompts):

- **Patient-to-Doctor:** Questions posed from the patient's perspective, reflecting confusion about a doctor's explanation or concerns about their condition. These are typically colloquial, lacking medical knowledge, and may carry emotional or binary (yes/no) undertones.
- **Doctor-to-Doctor:** Questions framed from a physician's perspective, emulating professional exchanges between doctors regarding specific aspects of a case's condition, diagnosis, or treatment. These focus on details from the chief complaint, imaging analysis, and clinical summary, and are presented in an open-ended discussion format.
- **Intern-to-Senior:** Simulating an intern consulting a senior physician on complex or challenging clinical issues within a case, often requiring reasoning. Questions primarily focus on the analysis of imaging results and are posed in an open-ended format.

**Implantation Details in MICS**   The MICS strategy leverages three key operations (Section 3.2) to iteratively search until the maximum depth is reached or a full-score reasoning path is identified. By applying MICS to complex clinical problems, we construct a set of high-quality, step-by-step reasoning data, thereby enabling MLLMs to learn reasoning progressively. Notably, to reduce data construction costs, we configure three mentor models [3, 23, 50] and six distinct intern models [3, 9] (three models, each with two different temperatures) to execute the search. Additionally, if the $(n + 1)$-th step is generated by mentor model $\theta$, $\theta$ can directly adopt the $(n + 2)$-th step from the complete solution generated in the previous search round, thereby avoiding redundant reasoning during the search process. Furthermore, we set the maximum search depth to $4$. Unlike mathematical problem-solving, which may require dozens of steps, resolving complex medical problems typically involves approximately $4$ to 7 steps, encompassing medical history analysis, image interpretation, differential diagnosis, and more. Given the limited number of steps, the basic reasoning logic is generally established by the time the maximum depth is reached. If the path still fails to enable interns to derive the correct answer, it indicates a very low-quality reasoning path, rendering further exploration unproductive. Finally, we perform quality control on the data from completed searches. Reasoning paths exhibiting characteristics such as "no upward trend", "consistently zero", or "rising then falling" are flagged as low-quality data (see Appendix B for details).

## B   Metrics

As the reasoning dataset in MMRP consists of open-ended questions, we not only employ DeepSeek-V3 [31] as the judge to determine the correctness of generated answers but also use the text embedding model "roberta-large" [36] to compute semantic similarity between the model's final answer and the ground truth. MICS-Score is used to evaluate the effectiveness of the reasoning path.

**Trend of Path Scores**   When using MICS to search for effective reasoning paths, we obtain a path score that records the highest MICS-Score of the reasoning step selected in each search iteration. Evidently, as the reasoning process deepens, this score should exhibit a gradually increasing trend. To evaluate the effectiveness of the MICS strategy, we analyze the trend of path score changes and

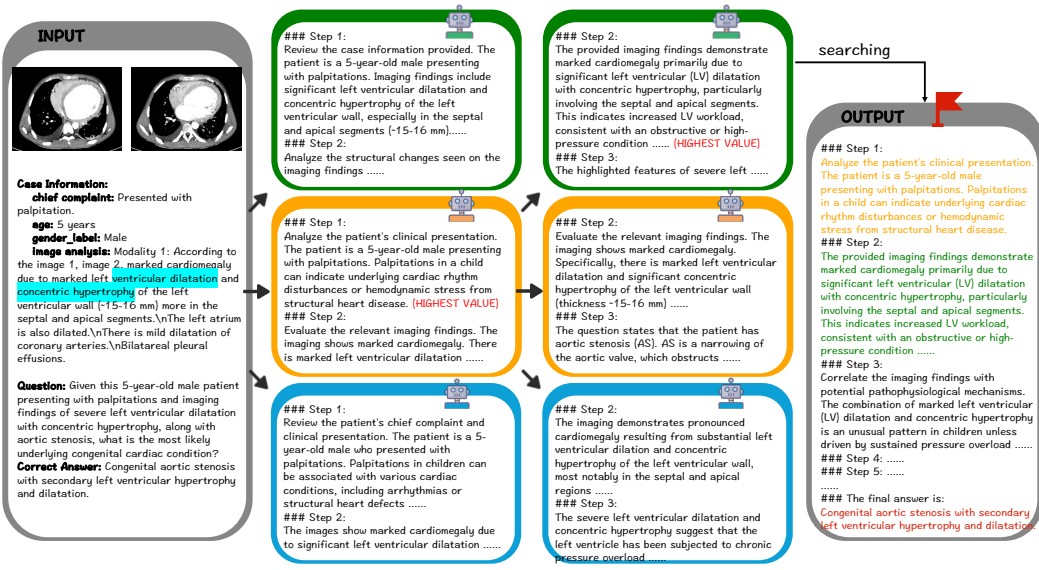

Figure 7: Qualitative illustration of effective medical reasoning paths search using MICS.

compare it with the vanilla method that does not employ MICS, with results presented in Figure 4 (a). We define four distinct trends: (1) Monotonically increasing, where effective reasoning paths are expected to show steadily increasing or at least stable scores. (2) Non-increasing, characterized by an overall downward trend. (3) Constant, where the path score remains unchanged. (4) Fluctuating, indicating unstable search with scores that vary unpredictably.

## C Training Sets

During the training process, we combine commonly-used medical VQA datasets with MMRP to train Chiron-o1. This approach aims to further enhance the model's visual understanding and question-answering capabilities, laying the groundwork for improved reasoning abilities. Results from ablation studies (Table 3) indicate that our configuration of the training set is reasonable and effective.

Table 4: The distribution of datasets used in each training stage. "HuatuoV_A" and "HuatuoV_I" refer to the Huatuo_PubMedVision_Alignment and Huatuo_PubMedVision_InstructionTuning VQA datasets, respectively.

| Dataset | Stage 1 | Stage 2 | Stage 3 |
|---|---|---|---|
| **MMRP Part 1** | 111260 | 55630 | 55630 |
| **MMRP Part 2** | – | 58780 | 29390 |
| **MMRP Part 3** | – | – | 183150 |
| **HuatuoV_A** | 129351 | 129351 | 646759 |
| **HuatuoV_I** | 129351 | 129351 | 646759 |
| **PMC_VQA** | 30520 | 30520 | 152603 |
| **VQA_RAD** | 1794 | 1794 | 8970 |
| **SLAKE** | 3951 | 3951 | 9835 |
| **PATH_VQA** | 3934 | 3934 | 19755 |

# D  Experiments

Since the MMMU(H&M) benchmark encompasses five distinct categories of medical questions, we compare Chiron-o1 with existing medical reasoning models, Med-R1 and MedVLM-R1, to analyze their performance across these subdomains. The results in Table 5 demonstrate that our model achieves substantial performance improvements across all categories of the test set. Notably, Chiron-o1 exhibits significant performance gains in "Clinical Medicine", "Diagnostics and Laboratory Medicine", and "Pharmacy", which focus heavily on clinical problems. This is attributed to the MMRP dataset, constructed based on complex clinical cases, enabling our model to demonstrate superior performance in addressing clinical problems.

Table 5: Results on the test set of MMMU(H&M). The subset is divided into five categories: BMS for Basic Medical Science, CM for Clinical Medicine, DLM for Diagnostics and Laboratory Medicine, P for Pharmacy, and PH for Public Health.

| Model | BMS | CM | DLM | P | PH | AVG |
|---|---|---|---|---|---|---|
| **Med-R1** | 38.1 | 32.7 | 27.8 | 29.1 | 33.7 | 32.7 |
| **MedVLM-R1** | 39.6 | 36.4 | 29.0 | 33.9 | 35.6 | 35.5 |
| **Chiron-o1-2B** | 42.9 | 42.5 | 42.0 | 44.1 | 39.1 | 42.1 |
| **Chiron-o1-8B** | 55.7 | 54.9 | 48.1 | 56.2 | 54.5 | 54.6 |

# E  Limitations

Our work introduces a novel reasoning path search strategy, MICS, to efficiently construct multimodal medical CoT data. Using the reasoning dataset established with MICS, we develop Chiron-o1, which exhibits robust visual understanding and reasoning capabilities. This may offer new insights and perspectives for multimodal reasoning in the medical domain. However, our approach has certain limitations to consider: (1) The MICS strategy requires collaboration between mentor models and intern models to search for reasoning paths, necessitating numerous costly API requests and valuable computational resources. (2) The multimodal reasoning dataset MMRP we proposed requires further expansion in scale, which will be a focus of our future work. Meanwhile, as medical CoT construction methods like MICS continue to advance, we hope the emergence of more reasoning datasets in the future.

# F Prompts

You are a medical education assistant tasked with creating a set of question-and-answer pairs (QA pairs) based on provided clinical case information. The case information includes the case title, the system involved (e.g., respiratory system, digestive system, etc.), the patient's chief complaint (including past medical history), the patient's gender, the patient's age, relevant imaging analysis results (potentially encompassing multiple modalities such as CT, MRI, X-ray, etc.), and the clinical summary and discussion. Please design the QA pairs in strict accordance with the following requirements:

- **Moderate Difficulty:** Questions and answers should be appropriate for the level of medical students or clinical interns, avoiding content that is overly complex or requires advanced clinical expertise.
- **Multiple-Choice Format:** Each question must offer 3-5 options, with only one correct answer and the remaining options serving as plausible but incorrect distractors.
- **Information Dependency:** Use only the information provided in the case; do not fabricate or introduce details beyond the case. If specific information (e.g., chief complaint, imaging analysis, or summary and discussion) is missing or unavailable, do not invent it and skip related content.
- **Logical Reasoning:** The correct answer must be logically deducible from the chief complaint, imaging analysis, or clinical summary and discussion within the case information.
- **Imaging Constraints and Handling:** Rely solely on the provided descriptions of imaging analysis results, which may involve multiple modalities (e.g., CT, MRI, etc.), and ensure differentiation between information from distinct modalities; do not involve direct access to or interpretation of raw clinical images.
- **QA Independence:** The wording of both questions and explanations must avoid using referential terms such as "this case", ensuring that each question is independent of the specific case context and can be understood and answered standalone.
- **Medical Terminology:** Employ precise, standardized medical terminology, maintaining a professional and accurate linguistic standard.

**Output Format:**
For each QA pair, provide:
- Question (concise, clear, and independently phrased)
- Options (labeled A, B, C, etc., including one correct answer)
- Correct Answer (clearly stated)
- Rationale (Summarize based on the provided information, using general medical reasoning logic to explain why the correct answer was chosen and distractors were ruled out, avoiding direct references to "the case" or specific context, and presenting an objective medical analysis).

- Generate 1/2/3 QA pair(s), ensuring that the questions address different aspects (e.g., chief complaint, imaging analysis, clinical discussion), while remaining strictly within the scope of the provided information.
- Please strictly follow the output format to generate QA pairs, and do not add the usual content. The form is as follows
"Question 1
Question: Which of the following imaging features best suggests that this patient has a high-grade brain tumor?
Options: A) Central cystic component with surrounding solid components
B) Diffusion restriction and contrast enhancement of the solid component
C) Homogenous enhancement without surrounding edema
D) Mildly elevated choline peak and Cho/Cr ratio
Correct Answer: B) Diffusion restriction and contrast enhancement of the solid component
Rationale: Diffusion restriction and contrast enhancement of the solid component are key features that suggest a high-grade tumor. Diffusion restriction usually indicates a high-cellular area, which is often associated with aggressive tumors such as high-grade gliomas. The presence of contrast enhancement further supports the possibility of a high-grade tumor. Option A is a feature of many brain tumors, but does not specifically indicate a high-grade tumor. Option C describes a more benign tumor appearance, while option D usually indicates a lower-grade tumor."

Figure 8: System prompt for generating QA pairs in Part 1 of MMRP.

Please construct QA pair(s) based on the following content, avoiding the use of referential phrases such as 'this case' or 'this image analysis'. Instead, describe the 'question' and 'rationale' in an objective manner, as if not referencing any specific case information. Please ensure that 'Question', 'Options', 'Correct Answer', and 'Rationale' are not followed by colons.
- Title: {title}
- System involved: {systems}
- Chief complaint: {presentation}
- Gender: {gender_label}
- Age: {age_label}
- Imaging analysis: {caption}
- Clinical summary and discussion: {discussion}

Figure 9: Input prompt for generating QA pairs in Part 1 of MMRP.

You are a medical education assistant tasked with generating a question-and-answer pair (QA pair) based on clinical case information provided by the user. The case information includes: case title, system involved, patient's chief complaint, patient's gender, patient's age, relevant imaging analysis results (which may include multiple modalities or sequences), and clinical summary and discussion. Please strictly adhere to the following requirements when designing the QA pair:

**Question Design Requirements:**
- The question must be posed from the perspective of a patient with a confirmed diagnosis, simulating their doubts about their condition, diagnosis, or treatment, and phrased as a yes/no question.
- The patient may lack medical knowledge, so the question should reflect their confusion, concerns, or skepticism about the doctor's explanations and treatment recommendations, potentially with emotional or challenging undertones.
- Appropriate forms of address may be included to reflect the clinical context.

**Answer Design Requirements:**
- The answer must be provided from the perspective of an experienced, professional doctor, based on logical reasoning from the case information—chief complaint, imaging analysis results, and clinical summary and discussion.
- The answer must be limited to "Yes" or "No", without providing explanations or reasoning.
- If the case information is insufficient to answer the question, respond with "Based on the current information, this cannot be determined," without fabricating or introducing details beyond the case.

**Notes:**
- Use only the information provided in the case; do not add fabricated details or external content.
- Rely solely on the described imaging analysis results, distinguishing between different modalities, without directly interpreting raw imaging data.

**Output Format:**
- Question: {question}
- Correct Answer: {answer}(limited to "Yes", "No", or "Based on the current information, this cannot be determined")

Figure 10: System prompt for constructing "Patient-to-Doctor" VQA data in Part 3 of MMRP.

You are a medical education assistant tasked with generating a question-and-answer pair (QA pair) based on clinical case information provided by the user. The case information includes: case title, system involved, patient's chief complaint, patient's gender, patient's age, relevant imaging analysis results (which may include multiple modalities or sequences), and clinical summary and discussion. Please strictly adhere to the following requirements when designing the QA pair:

**Question Design Requirements:**
- The question must be posed from the perspective of a doctor, simulating professional discussion between physicians about a specific aspect of the case related to the condition, diagnosis, or treatment.
- The question should reflect a doctor's expertise, focusing on details from the chief complaint, imaging analysis results, and clinical summary, and be phrased as an open-ended discussion question (not a yes/no question).
- Appropriate forms of address may be included to reflect the professional context.

**Answer Design Requirements:**
- The answer must be provided from the perspective of an experienced, professional doctor, based on analysis of the case information—chief complaint, imaging analysis results, and clinical summary and discussion.
- The answer must consist of a single medical term or a concise professional summary, without detailed explanation.

**Notes:**
- Use only the information provided in the case; do not add fabricated details or external content.
- Rely solely on the described imaging analysis results, distinguishing between different modalities, without directly interpreting raw imaging data.

**Output Format:**
- Question: {question}
- Correct Answer: {answer}

Figure 11: System prompt for constructing "Doctor-to-Doctor" VQA data in Part 3 of MMRP.

You are a medical education assistant tasked with generating a question-and-answer pair (QA pair) based on clinical case information provided by the user. The case information includes: case title, system involved, patient's chief complaint, patient's gender, patient's age, relevant imaging analysis results (which may include multiple modalities or sequences), and clinical summary and discussion. Please strictly adhere to the following requirements when designing the QA pair:

**Question Design Requirements:**
- The question must be posed from the perspective of a clinical intern, simulating a intern consulting a senior doctor about a complex or challenging clinical issue in the case, typically requiring reasoning.
- The question should reflect the intern's learning needs and exploration of complex knowledge, primarily focusing on the analysis of imaging results, and be phrased as an open-ended question.
- Appropriate forms of address (e.g., "Teacher" or "Professor") may be included to reflect the educational context.

**Answer Design Requirements:**
- The answer must be provided from the perspective of an experienced senior doctor, based on logical reasoning from the case information—chief complaint, imaging analysis results, and clinical summary and discussion—addressing and explaining the viewpoint.
- The answer must consist of a single medical term or a concise professional summary, without detailed explanation.

**Notes:**
- Use only the information provided in the case; do not add fabricated details or external content.
- Rely solely on the described imaging analysis results, distinguishing between different modalities, without directly interpreting raw imaging data.

**Output Format:**
- Question: {question}
- Correct Answer: {answer}

Figure 12: System prompt for constructing "Intern-to-Senior" VQA data in Part 3 of MMRP.

Please construct a QA pair based on the following content.
- Title: {title}
- System involved: {systems}
- Chief complaint: {presentation}
- Gender: {gender_label}
- Age: {age_label}
- Imaging analysis: {caption}
- Clinical summary and discussion: {discussion}

Figure 13: Input prompt for constructing VQA data in Part 3 of MMRP.

Using the provided medical images and partial thought process, deduce the correct answer of the question through rigorous reasoning. Ensure the response is concise, accurate, and conforms to medical terminology standards. Provide only the final answer.

Format your response with the following format:
### The final answer is:

Question: {question}
Reasoning prefix: {reasoning_prefix}

Figure 14: Prompt for the intern model to perform reasoning based on the mentor model's guidance.

Evaluate whether the model's answer aligns with the correct answer semantically. Output 'Yes' only if the model's answer matches the correct result, and 'No' if it does not match or if the correctness is unclear. Provide only 'Yes' or 'No' as the output, without any explanation.

Question: {question}
Model's answer: {model_answer}
Correct answer: {gt_answer}

Figure 15: Prompt for comparing the final answer produced by the intern model with the ground truth.

Given a specific question about the images, the patient's case information (such as age, gender, chief complaint and some relevant image analysis ), your goal is to generate a detailed, step-by-step thought process that leads to the correct answer.
1. Your thought process must rely solely on the provided information. Do not fabricate details or introduce information not present in the inputs.
2. Approach the task as if the answer is unknown, avoiding any shortcuts or assumptions that the gold standard answer is already understood.
3. If the thought process involves observations related to images, present those observations as if they were directly derived from the images themselves, without referencing image analysis.
4. Adapt your thought process to the complexity of each case, using fewer reasoning steps for simpler problems and more thorough analysis for complex ones, mirroring the flexible and analytical mindset of a skilled clinician.

Format your response with the following format:
### Step 1:
### Step 2:
...
### The final answer is:

Case Information: {case_info}
Question: {question}
Correct Answer: {gt_answer}
Reasoning prefix: {reasoning_prefix}

Continue your response by building directly on the existing thought process, ensuring that no changes are made to the content of the prior reasoning, and provide a clear and coherent completion of your answer. Just output the completed part.

Figure 16: Prompt for the mentor model to exploring reasoning paths based on prefix.

# G  Qualitative Analysis of Medical Reasoning Models

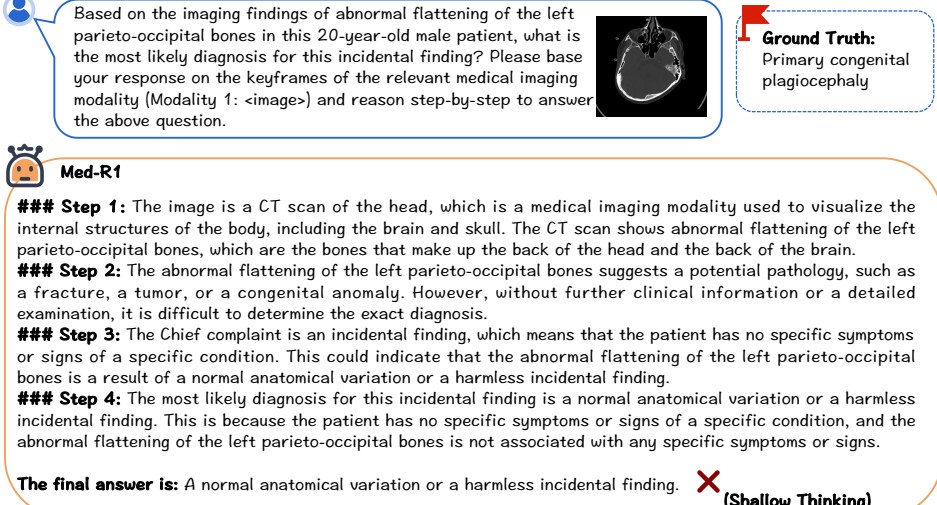

Figure 17: Reasoning result of Med-R1.

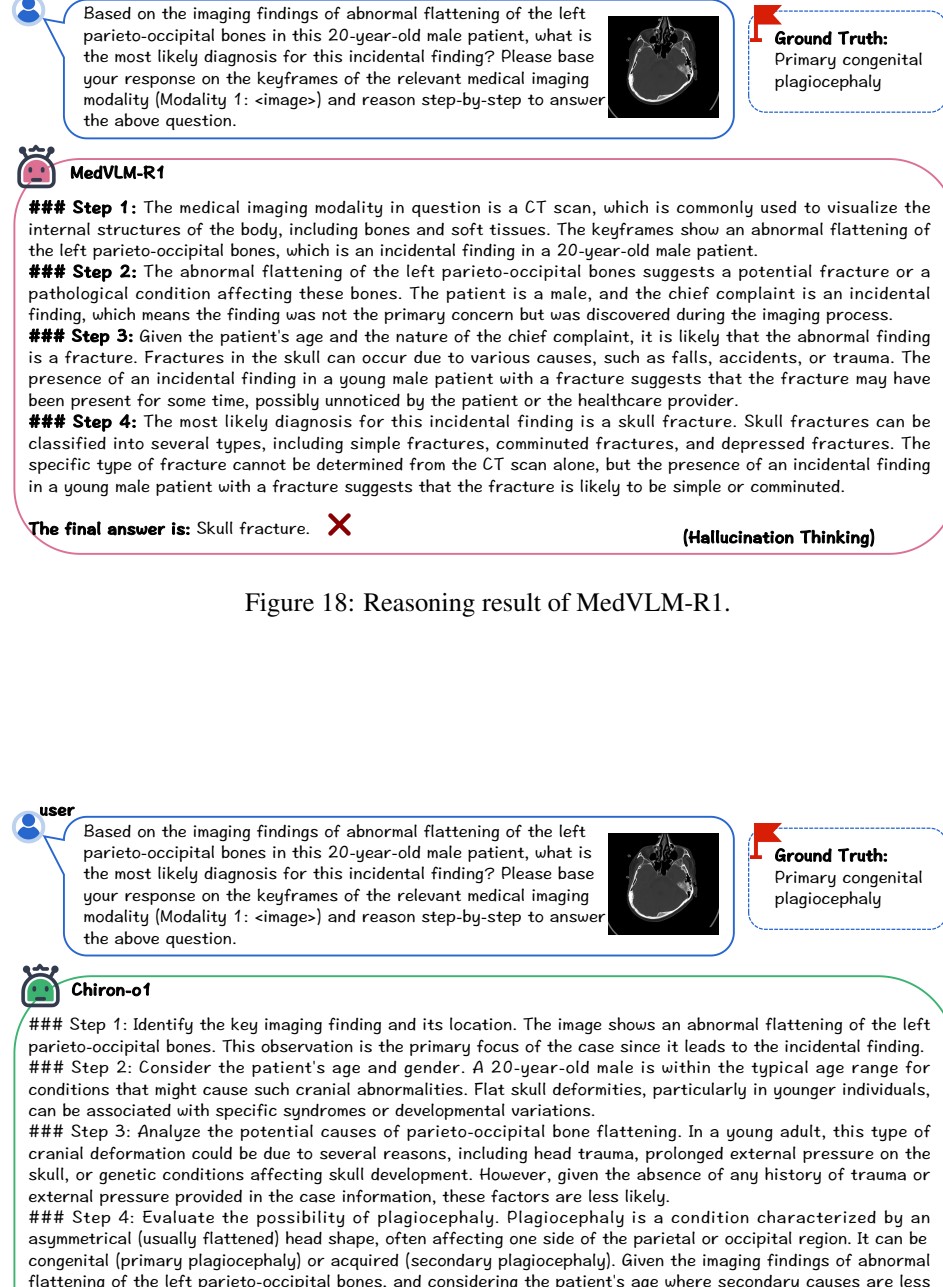

Figure 18: Reasoning result of MedVLM-R1.

Figure 19: Reasoning result of Chiron-o1.

# H Case Study

**Question:** Doctor, I've had this pain in my finger for a year now, and you're telling me it's because of a tiny bone tumor? Are you sure it's not just arthritis or something less serious?

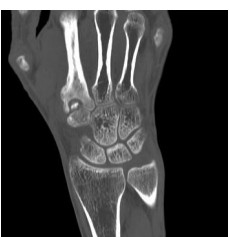

**Reasoning:**
Let's think about how to solve this problem clearly and reasonably step by step.
### Step 1: Consider the patient's chief complaint and history. The patient is a 30-year-old male who has experienced pain in his right index finger for one year without any history of trauma. This information suggests that the cause of the pain is likely not due to acute injury but rather a chronic condition.
### Step 2:Analyze the provided image findings. The image reveals a small, well-defined, lobulated lesion measuring approximately 4.4 x 3.4 x 3.3 mm along the dorsal aspect of the 2nd metacarpal bone, near the level of the waist of the bone and the base of the 2nd proximal phalanx. The lesion appears to be exophytic, meaning it grows outward from the bone surface. There is no evidence of pathologic fracture, which indicates that the lesion, while potentially aggressive, has not yet compromised the structural integrity of the bone significantly.
### Step 3:Evaluate the characteristics of the lesion. The size and appearance of the lesion are consistent with a small bone tumor rather than a common degenerative condition like arthritis. Arthritis typically presents with joint space narrowing, osteophyte formation, and possibly synovial inflammation, none of which are mentioned in the image findings. The absence of these features supports the diagnosis of a bone tumor over arthritis.
### Step 4: Consider the differential diagnosis for a small exophytic bone lesion in this location. Given the patient's age and the characteristics of the lesion, possibilities include osteochondroma, enchondroma, or other benign bone tumors. The lack of symptoms suggesting infection or malignancy (such as rapid growth, systemic symptoms, or pathological fractures) further supports a benign etiology.
### Step 5:Assess the certainty of the diagnosis. While the imaging findings strongly suggest a small bone tumor, it is important to acknowledge that definitive diagnosis may require additional tests such as biopsy or follow-up imaging to monitor any changes in the lesion's size or behavior. However, based on the current information, the evidence points more towards a bone tumor than a less serious condition like arthritis.

**Final answer:** Yes, based on the imaging findings and the patient's clinical presentation, it is reasonable to conclude that the pain is likely due to a small bone tumor rather than a less serious condition like arthritis.

Figure 20: Chiron-o1 Reasoning on Example 1.

**Patient-to-Doctor (Modality: MRI)**

**Question:** Doctor, I just found out I have these cysts in my head, but I feel fine. Are you sure I don't need surgery or any treatment right now?

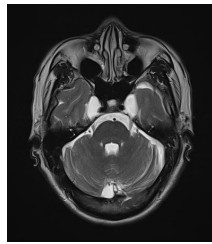

**Reasoning:**

Let's think about how to solve this problem clearly and reasonably step by step.

### Step 1: Analyze the patient's clinical presentation.The patient is a 50-year-old male in whom the finding of cystic structures in the cerebellopontine (CP) cisterns was incidental. The patient reports feeling fine and has no specific symptoms related to these findings.

### Step 2: Review the imaging findings described.The MRI shows bilateral cystic structures in the CP cisterns, adjacent to the inner table of the petrous bones. They follow cerebrospinal fluid (CSF) signal on all sequences, indicating they are likely filled with CSF. They are described as non-enhancing and extra-axial.

### Step 3: Interpret the significance of the imaging findings in the context of clinical practice. The cystic structures are described as bilateral, non-enhancing, extra-axial, and CSF-signal intensity, which strongly suggests they are arachnoid cysts. Arachnoid cysts are benign, CSF-filled sacs that develop between the brain and the arachnoid membrane. When small and asymptomatic—such as in this case where the patient reports no symptoms—they often do not require intervention.

### Step 4: Consider the natural history and management options. Most small, uncomplicated arachnoid cysts managed conservatively with observation, as there are no signs of mass effect, hydrocephalus, or associated symptoms. The absence of symptoms, normal brain parenchyma, and the typical CSF signal intensity of the lesions support a benign course. Intervention (e.g., surgery or endoscopic drainage) is generally reserved for cases with symptoms (e.g., headache, hearing loss, cranial nerve dysfunction) or complications (e.g., obstructive hydrocephalus), none of which are present here.

### Step 5: Weigh the risks and benefits of intervention in this specific patient. Given the lack of symptoms, the incidental nature of the finding, and the benign imaging features, the risks of unnecessary surgery (e.g., infection, bleeding, cerebrospinal fluid leak) outweigh the benefits in this asymptomatic patient. Therefore, conservative management with periodic clinical and imaging follow-up is appropriate.

**Final answer:** No.

Figure 21: Chiron-o1 Reasoning on Example 2.

**Doctor-to-Doctor (Modality: MRI)**

**Question:** Based on the MRI findings of a solid-cystic lesion in the left cerebellar hemisphere with intense enhancement of the solid component, a feeding vessel, and mass effect compressing surrounding structures, what is the most likely diagnosis in this 40-year-old male presenting with giddiness?

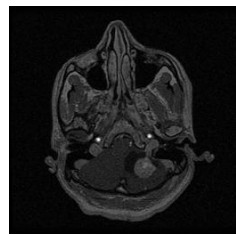

**Reasoning:**
Let's think about how to solve this problem clearly and reasonably step by step.
### Step 1: Consider the patient's chief complaint and location of the lesion. The patient is a 40-year-old male presenting with giddiness, and the MRI reveals a space-occupying lesion in the left cerebellar hemisphere. Given that the cerebellum is involved, this finding could explain the symptoms as it plays a crucial role in balance and coordination.
### Step 2: Analyze the characteristics of the lesion. The lesion is described as solid-cystic in nature with intense enhancement of the solid component on post-contrast T1 imaging. This pattern of enhancement suggests a vascular tumor, as the feeding vessel noted further supports this. The presence of both solid and cystic components narrows down the differential diagnosis to tumors that can exhibit such features.
### Step 3: Evaluate the anatomical and functional implications of the lesion. The lesion is causing significant mass effect on the fourth ventricle, leading to its shift to the right. This obstruction of cerebrospinal fluid (CSF) flow can result in obstructive hydrocephalus, which is a known cause of giddiness and other neurological symptoms. Additionally, the lesion is compressing the left middle cerebellar peduncle and the cerebellar vermis, further supporting its impact on cerebellar function.
### Step 4: Consider the differential diagnosis for a solid-cystic enhancing lesion in the cerebellum. Common considerations include hemangioblastoma, metastasis, ependymoma, or other less common tumors. However, the intense enhancement of the solid component and the presence of a feeding vessel are highly suggestive of a hemangioblastoma. Hemangioblastomas are known to have a rich vascular supply, which explains the observed enhancement and the presence of a feeding vessel. They also frequently occur in the cerebellum and can present with symptoms related to mass effect and hydrocephalus.
### Step 5: Correlate the imaging findings with the clinical presentation. The patient's giddiness is consistent with the location and size of the lesion in the cerebellum, which is critical for maintaining balance and coordination. The combination of the imaging characteristics and the clinical symptoms strongly points towards a specific diagnosis.

**Final answer:** Cerebellar hemangioblastoma.

Figure 22: Chiron-o1 Reasoning on Example 3.

**Doctor-to-Doctor (Modality: Ultrasound)**

**Question:** Given this 35-year-old female with a history of cesarean section presenting with 6 weeks amenorrhea, positive pregnancy test, and low abdominal pain, along with ultrasound findings of twin gestational sacs implanted in the thinned cesarean scar without yolk sacs or embryos, what is the most appropriate management to prevent catastrophic uterine rupture?

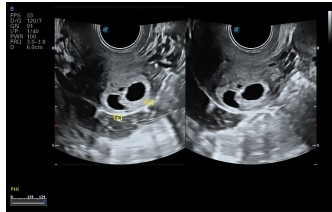

**Reasoning:**

Let's think about how to solve this problem clearly and reasonably step by step.

### Step 1: Review the patient's history and current presentation. The patient is a 35-year-old female with a history of a cesarean section 12 years ago, now presenting with 6 weeks of amenorrhea, a positive pregnancy test, and low abdominal pain. This background suggests an increased risk for complications related to her previous surgery.

### Step 2: Analyze the ultrasound findings. The image reveals two gestational sacs located in the lower uterus at the site of the thinned cesarean section scar. There are no yolk sacs or embryos visible within these sacs at this early stage of pregnancy. The presence of gestational sacs without yolk sacs or embryos could indicate early-stage pregnancies or other abnormal implantations. The location of these sacs at the cesarean scar is particularly concerning as it places them in a high-risk area for uterine rupture.

### Step 3: Consider the implications of the findings. The combination of the patient's history, symptoms, and ultrasound results strongly suggests a diagnosis of cesarean scar pregnancy. This condition is characterized by the implantation of the fertilized egg at the site of the previous cesarean incision, which can lead to catastrophic uterine rupture if left untreated. The absence of fetal cardiac activity further supports the severity of the situation, as it indicates that the pregnancies may not have a viable future.

### Step 4: Evaluate management options. Given the high risk of uterine rupture and potential for life-threatening hemorrhage, immediate intervention is necessary to prevent complications. Conservative management would not be appropriate due to the imminent risk of rupture. Surgical options such as hysteroscopic excision or cesarean scar curettage are viable choices to remove the gestational tissue and prevent further complications.

### Step 5: Determine the most appropriate management. Considering the patient's age, the high risk associated with cesarean scar pregnancies, and the potential for severe complications if left untreated, hysteroscopic excision or cesarean scar curettage emerges as the most suitable course of action. These procedures aim to safely remove the gestational tissue while minimizing the risk of uterine damage and subsequent fertility issues.

**Final answer:** Hysteroscopic excision or cesarean scar curettage (high-risk cesarean scar pregnancy).

Figure 23: Chiron-o1 Reasoning on Example 4.

**Intern-to-Senior (Modality: DSA)**

**Question:** Teacher, based on the imaging analysis showing occlusion of the left anterior descending coronary artery at its origin and successful stent placement, what is the most likely diagnosis for this patient's chest pain?

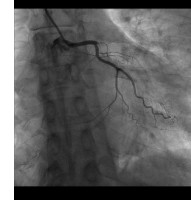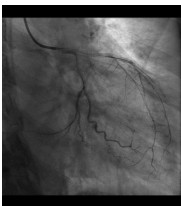

**Reasoning:**
Let's think about how to solve this problem clearly and reasonably step by step.
### Step 1: Analyze the patient's presentation and demographics.The patient is a 65-year-old male presenting with chest pain. Chest pain in this age group and gender raises a high suspicion for cardiovascular pathology, particularly myocardial infarction or ischemia.
### Step 2: Evaluate the findings from the coronary angiography (Modality 1).The angiography reveals specific abnormalities in the left coronary artery system. There is an occlusion, meaning a complete blockage, noted at the origin of the left anterior descending (LAD) artery. Concurrently, there is mild narrowing observed in the first septal branch of the left anterior descending artery. The other major coronary arteries examined (circumflex and left main) appear normal.
### Step 3: Define the condition based on the angiographic findings.An occlusion representing a complete blockage of blood flow within a coronary artery is termed occlusive coronary artery disease. The location of the occlusion is critical for determining the specific type.
### Step 4: Correlate the specific findings with known patterns of occlusive coronary artery disease.The LAD artery gives rise to septal branches. An occlusion located precisely at the origin of the LAD is known as an occlusive septal branch coronary occlusion (O-SBCO) or more specifically, an occlusive proximal LAD occlusion. This specific location directly explains the described \"occlusion of the left anterior descending coronary artery at its origin\" and the subsequent \"mild narrowing of the first septal branch\".
### Step 5: Incorporate the procedural information.The presence of stent placement confirms the clinical significance of this occlusion and the diagnosis of occlusive coronary artery disease. The fact that perfusion was restored (implied by the need for stenting) explains the patient's chest pain, as the LAD supplies a significant portion of the anterior myocardium.
### Step 6: Synthesize the findings into a final diagnosis.Combining the clinical presentation (chest pain) with the specific angiographic findings (occlusion at the origin of the LAD, mild distal narrowing, successful stenting), the most accurate and specific diagnosis describing the coronary artery occlusion is an Occlusive Proximal LAD Occlusion, which is a type of occlusive coronary artery disease.

**Final answer:** Occlusive proximal left anterior descending (LAD) occlusion.

Figure 24: Chiron-o1 Reasoning on Example 5.

**Intern-to-Senior (Modality: X-ray)**

**Question:** Teacher, based on the imaging findings of a calcified focal area lateral to the greater tubercle in this 45-year-old male with chronic left shoulder pain, what is the most likely diagnosis?

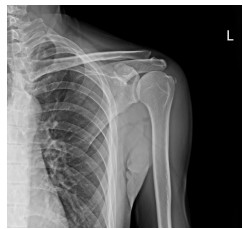

**Reasoning:**

Let's think about how to solve this problem clearly and reasonably step by step.

### Step 1: Consider the patient's chief complaint and demographic information. The patient is a 45-year-old male presenting with chronic left shoulder pain. Chronic shoulder pain in this age group can be caused by various conditions, including rotator cuff pathology, osteoarthritis, or less commonly, calcific tendinitis.

### Step 2:Analyze the imaging findings provided. The images reveal a calcified focal area lateral to the greater tubercle. This specific location is highly suggestive of calcific tendinitis, as the greater tubercle is where the supraspinatus tendon attaches. Calcific tendinitis is characterized by the formation of calcium deposits within the tendon, which can lead to inflammation and chronic pain.

### Step 3:Evaluate the clinical correlation. Given the patient's age and the chronic nature of his symptoms, combined with the imaging findings of a calcified area near the greater tubercle, calcific tendinitis emerges as the most likely diagnosis. Other conditions such as adhesive capsulitis or degenerative changes would not typically present with a focal calcified area in this specific location.

### Step 4:Rule out alternative diagnoses. While conditions like osteoarthritis or rotator cuff tears can cause shoulder pain, they usually do not present with a focal calcified area on imaging. Osteoarthritis would more likely show joint space narrowing or osteophytes, and rotator cuff tears would demonstrate discontinuity or retraction of the tendon, neither of which are described here.

**Final answer:** Calcific tendinitis of the supraspinatus tendon.

Figure 25: Chiron-o1 Reasoning on Example 6.

