# OpenReview forum: "Chiron-o1: Igniting Multimodal Large Language Models towards Generalizable Medical Reasoning via Mentor-Intern Collaborative Search"
_NeurIPS.cc/2025/Conference — NeurIPS 2025 poster_

### Official Review · Reviewer_TAs2 · 2025-06-05

**Clarity:** 3
**Significance:** 2
**Originality:** 2
**Rating:** 4
**Confidence:** 4

**Summary:**

This paper proposes a novel approach for searching and evaluating effective reasoning paths towards critical diagnosis in the medical domain. The authors construct a multi-task medical reasoning dataset with ranked difficulty and a new medical MLLM devised via a curriculum learning strategy. Extensive experiments demonstrate that Chiron-o1 achieves state-of-the-art performance across a list of medical visual question answering and reasoning benchmarks.

**Questions:**

I wonder if the multi-stage learning paradigm is generalizable across different MLLMs. The ablations about this is only conducted on MMMU with Chiron-o1-8B.

**Ethical Concerns:**

["NO or VERY MINOR ethics concerns only"]

**Final Justification:**

The clarifications regarding the dataset construction, image resolution, and the design rationale behind the multi-stage training strategy somewhat addressed my concerns. The additional experimental results are helpful. Considering the data is not from real clinical cases and I already gave a positive rating, I will keep my original scores.

**Limitations:**

Yes

**Paper Formatting Concerns:**

- The figure 4 is too small.
- There is too much margin on the left and right hand side of figure 1.

**Quality:**

3

**Strengths And Weaknesses:**

Strengths:

- A multimodal medical dataset comprising QA pairs is introduced, which can contribute to the medical community.
- The authors propose a new multimodal medical model that demonstrates outstanding reasoning abilities in handling both in-domain and out-of-domain complex clinical problems.
- The proposed approach achieves competitive performance compared to previous SOTA medical MLLMs through extensive experiments across multiple benchmarks.

Weaknesses:

- The proposed medical dataset is basically a reorganization of existing medical datasets. There is no new high-quality dataset annotated in this paper. The contribution is limited.
- As shown in Figure 5, the dataset is imbalanced across different body systems.
- The source and resolution of the images in datasets are different and imbalanced. Also there is no high-resolution images, limiting the usefulness of the proposed dataset
- The novelty of the Multi-Stage SFT is limited. It is basically the curriculum learning pattern. There is also no ablation and experiment results supporting that it is better than every single single-stage learning.
- More qualitative results is needed for different test set.

---

> ### Author Rebuttal · Authors · 2025-07-31
>
> Thank you for your valuable feedback. We’re thrilled that the reviewer recognizes the “outstanding reasoning abilities” of our new multimodal medical model and the “competitive performance” of our approach, as well as the contribution of our multimodal medical dataset. Please see our response to the reviewer’s concerns below.
>
> **Re weakness 1:** Thank you for your concern. We would like to kindly clarify that the metadata for the MMRP dataset is sourced from the Radiopaedia public teaching website, and extracting valuable real clinical case data from it is a labor-intensive process. Based on the collected data, we proposed a novel CoT data construction method, MICS, as acknowledged by Reviewers eXAH and kkRL. This method enables collaborative search between mentor and intern models to identify high-quality reasoning paths, thereby completing the annotation of the reasoning process for the entire dataset. As you noted, manual data annotation is currently a time-consuming and labor-intensive task, not only in the medical field but also in other domains. Therefore, we aim to propose an automated CoT data annotation process to alleviate this issue and further enhance the performance of MLLMs in medical reasoning. To ensure that the constructed data adheres to strict clinical logic and practical value, we recruited three experienced radiologists to review and validate the MMRP dataset, retaining only high-quality CoT annotations for subsequent training.
>
> **Re weakness 2:** Thank you for your critical and insightful feedback. We had previously noted the issue you raised. The imbalanced data distribution in MMRP is inherited from the varying patient volumes across clinical departments. However, as shown in the experimental results (Table 1 and Table 2) in the manuscript, the existing data is sufficient to improve the model’s VQA and reasoning performance. Following your suggestion, we plan to expand the MMRP dataset in the future, balancing data across different body parts and focusing on collecting cases of rare diseases.
>
> **Re weakness 3:** Thank you for your constructive critique. Indeed, the images in our dataset are sourced from real data uploaded by doctors from medical centers in different countries. We believe this enhances the diversity of the MMRP dataset, thereby improving the broad applicability of MICS and the generalizability of Chiron-o1. As you pointed out, variations in resolution and data sources may introduce biases. To address this, we normalized the resolution of all images before using MICS for data construction. Additionally, our analysis shows that the maximum image resolution in the MMRP dataset is 3001 × 6229, with an average resolution of 660.5 × 676.8. The table below presents the pixel count distribution of all images in the MMRP dataset.
>
> | Resolution range (total number of pixels) | Numbers  |
> |:-----------------------------------------:|:--------:|
> | <1k                                       | 2        |
> | 1k-10k                                    | 905      |
> | 10k-50k                                   | 6,865    |
> | 50k-100k                                  | 827      |
> | 100k-200k                                 | 578      |
> | 200k-500k                                 | 536      |
> | 500k-1000k                                | 175      |
> | >1000k                                    | 49       |
>
> **Re weakness 4:** Thank you for your valuable perspective. We would like to kindly clarify that MICS, used for annotating high-quality CoT data, and MICS-Score, used as a metric to evaluate reasoning paths, are the primary contributions of this paper. Building on the MMRP dataset, Chiron-o1, trained through a three-stage SFT process, demonstrates robust reasoning performance. As you mentioned, the stage-wise training approach is essentially a form of curriculum learning. However, we designed the stages to emulate the learning logic of real medical students. Specifically, stage 1 uses a large volume of QA data to enhance the model’s ability to address text-only problems; stage 2 leverages real images and their annotations to improve image-text alignment; and stage 3 further strengthens reasoning performance using high-quality CoT data annotated by MICS. To demonstrate the advantage of this strategy over single-stage training, the results presented in the table below show that the three-stage SFT approach indeed leads to better model performance. In the future, we plan to incorporate RL algorithms (such as GRPO and PPO) into the SFT training framework to further enhance the generalizability of reasoning capabilities.
>
> | Stage | VQA-RAD | SLAKE | MMMU(H&M) | MedXpertQA_MM |
> |-----------|:-------:|:-----:|:---------:|:------------:|
> | **Stage 1** | 76.1    | 82.5  | 47.7      | 23.7         |
> | **Stage 1 + 2** | 76.1   | 82.7  | 48.4      | 23.2         |
> | **Stage 3** | 75.3    | 81.2  | 51.1      | 23.7         |
> | **Stage 1 + 2 + 3** | 76.8   | 83.2  | 54.6      | 24.2         |
>
> **Re weakness 5:** Thank you for your constructive critique. Your suggestions are highly valuable for improving the quality of our paper. Following your and other reviewers’ recommendations, we conducted further investigations into several questions, including whether MLLMs genuinely utilize image features for reasoning, the distribution of failure cases in data types constructed by MICS, and whether incorrect guidance in input information negatively impacts model reasoning. These qualitative experimental results will be included in the revised manuscript. Additionally, we plan to collect data from more diverse sources to further validate the model’s generalizability.
>
> **Re question 1:** Thank you for highlighting this issue. As shown in Figure 4(b) of the paper, we conducted ablation studies on multiple benchmarks, including VQA-RAD, SLAKE, MMMU(H&M), and MedXpertQA_MM, to demonstrate the effectiveness of multi-stage training for Chiron-o1-8B. To further verify that the multi-stage SFT paradigm can generalize to other MLLMs, we extended our investigation to Chiron-o1-2B. The specific results, shown in the table below, clearly indicate that the multi-stage training strategy also improves VQA and reasoning performance for other MLLMs. This success is attributed to the design of each stage, which is inspired by the learning process of real doctors.
>
> | Stage | VQA-RAD | SLAKE | MMMU(H&M) | MedXpertQA_MM |
> |-----------|:-------:|:-----:|:---------:|:------------:|
> | **Stage 1** | 71.3    | 81  | 37.5      | 21.2         |
> | **Stage 1 + 2** | 69.5   | 82.9  | 36      | 21         |
> | **Stage 3** | 68.8    | 71.2  | 35.5      | 21.1         |
> | **Stage 1 + 2 + 3** | 75.4   | 85.3  | 42.1      | 21.5         |
>
> **Re paper formatting concerns:** Thank you for your kind reminds. We will revise the figures in the updated manuscript.

---

### Official Review · Reviewer_kRLm · 2025-06-29

**Clarity:** 3
**Significance:** 3
**Originality:** 3
**Rating:** 4
**Confidence:** 4

**Summary:**

Chain-of-Thought (CoT) data is crucial for enhancing the reasoning capabilities of medical multi-modal large language models (MLLMs). The authors propose a Mentor-Intern Collaborative Search (MICS) method, which generates optimal CoT data by evaluating the scores of different reasoning paths. Based on this approach, they construct the MMRP dataset and train the Chiron-o1 model, which achieves outstanding performance on multiple VQA and reasoning benchmarks.

**Questions:**

1. Is it possible for the dataset constructed in this paper to be open-sourced? Making it publicly available would be a valuable contribution to the entire research community.
2. Medical imaging is an inherently rigorous discipline, and the capabilities of general-purpose MLLMs are limited in medical contexts. While optimizing the MLLM output distribution using certain metrics (such as the MICS-score) can to some extent improve the quality of generated CoT data, can CoT data generated without leveraging dedicated medical knowledge or tools like knowledge graphs truly be considered reliable? How can the correctness of the underlying medical logic be ensured? I hope the authors can provide a response on this issue.
3. The authors need to demonstrate that the proposed MLLM does in fact incorporate imaging information in its output, rather than relying solely on prompts and textual knowledge the model has acquired.

**Ethical Concerns:**

["NO or VERY MINOR ethics concerns only"]

**Final Justification:**

The additional discussion and experiments have largely addressed my main concerns. I encourage the authors to further clarify these points in the final version. Hence, I recommend accepting this paper.

**Limitations:**

yes

**Quality:**

3

**Strengths And Weaknesses:**

Strengths:
1. The experiments in the paper are comprehensive, the structure is clear, and the writing is well-organized.
2. The Mentor-Intern Collaborative Search (MICS) strategy is straightforward and plays an important role in improving the quality of CoT data.

Weaknesses:
1. MICS relies on the intern's feedback to refine the CoT paths generated by the mentor; however, the method essentially depends on the mentor model's capability to produce CoT data. Is the mentor model itself sufficiently competent to generate reliable and high-quality reasoning data? This point warrants further discussion.
2. The paper presents several examples (e.g., Figure 3, Figure 20) where the prompts provide extensive sign-related information, which is almost equivalent to revealing the answer. The reviewer is unsure whether the model is genuinely leveraging image features, or is simply generating responses based on the content of the prompt. The authors should validate this.
3. In many real clinical scenarios, such as medical screening or physical examinations, there is often no sign-specific prompt and only imaging data is available—the model needs to generate a diagnostic report. How does the model perform under such conditions?
4. If the prompt contains incorrect sign/disease-related information (e.g., the patient reports abdominal pain and is assumed to have appendicitis, but actually has a bowel obstruction), will the model be misled and provide suggestions along the wrong path? The authors should investigate and verify this.

---

> ### Author Rebuttal · Authors · 2025-07-31
>
> Thank you for your thoughtful review. We’re pleased that the reviewer finds the Mentor-Intern Collaborative Search (MICS) strategy “straightforward” and impactful for improving CoT data quality, and appreciates the “comprehensive” experiments and “well-organized” writing. Please see our response to the reviewer’s concerns below.
>
> **Re weakness 1:** Thank you for your valuable and detailed comments. Currently, the majority of CoT data construction methods are derived through distillation from closed-source models. We would like to kindly clarify that using large-scale models as mentor models to construct reasoning paths is a well-established approach. Building on this, our proposed MICS effectively enables collaborative search between mentor and intern models to ensure CoT quality. To ensure the reliability and validity of the MMRP_Part3 dataset, we recruited three doctoral students with practical clinical experience in radiology to manually review and validate the data, ultimately filtering out dozens of low-quality or non-practical samples. As evident from Table 1, Table 2, and the case studies in the manuscript, post-training with MICS-constructed data significantly enhances Chiron-o1’s VQA and reasoning capabilities. Furthermore, we tested the direct distillation CoT construction method against MICS using MICS-Score. The results, which show a gradual increase in MICS-Score as reasoning steps deepen, suggest that our approach is effective. The results presented in the table below demonstrate that direct distillation using the mentor model (“vanilla”) can yield a certain proportion of high-quality CoT data, and employing MICS further increases this proportion.
>
> | Methods   | valid (%) | non-increasing (%) | constant (%) | fluctuating (%) |
> |:---------:|:---------:|:------------------:|:------------:|:---------------:|
> | **vanilla** | 65.0      | 8.3                | 14.7         | 12.0            |
> | **MICS**    | 74.7      | 4.0                | 12.3         | 9.0             |
>
> **Re weakness 2:** Thank you for your helpful comment on this point. In fact, the metadata used in MMRP originates from real clinical cases uploaded by multiple medical centers, enriched with detailed clinical annotations (e.g., imaging interpretations and case discussions). We would like to kindly clarify that we only utilized this additional information to construct high-quality QA and CoT, aiming to minimize hallucinations. During the inference stage, some information (e.g., detailed imaging findings) is provided as input, but this is insufficient to reveal the final answer and is intended solely to assist the model in interpreting images. For instance, the “abnormal flattening of the left parieto-occipital bones” mentioned in Figure 3 does not equate to the ground truth “Primary congenital plagiocephaly.” To further validate whether the model genuinely leverages image features, we conducted experiments on the case in Figure 3, allowing the model to reason with and without images. In the latter case, the model produced an incorrect answer, “Flattened parieto-occipital bones, a normal anatomical variant,” and introduced more hallucinations due to the absence of imaging data. In the revised manuscript, we will include the reasoning processes for both scenarios to provide readers with a clearer understanding of this point.
>
> **Re weakness 3:** Thank you for your suggestion to refine this aspect. The report generation tasks you mentioned (e.g., physical examination reports, chest X-ray reports) are indeed common in real-world medical scenarios. Honestly, although our training data did not include this type of multimodal data, we believe Chiron-o1 can effectively generalize to out-of-domain tasks. Specifically, we constructed a dataset based on MIMIC-CXR for chest X-ray report generation, encompassing two tasks: finding generation and impression generation. We then evaluated Chiron-o1’s report generation performance using BERTScore, BLEU, ROUGE-1, ROUGE-2, and ROUGE-L as metrics. The evaluation results for these two tasks are presented in Tables 1 and 2 below. It is evident that, compared to general large-scale MLLMs and foundation models specialized for chest X-rays, Chiron-o1 achieves comparable performance.
>
> |     Models (Finding Generation)             | BERTScore | BLEU  | ROUGE-1 | ROUGE-2 | ROUGE-L |
> |-------------------------|:---------:|:-----:|:-------:|:-------:|:-------:|
> | **HuatuoGPT-Vision-7B**   | 85.4      | 9.7   | 27.8    | 5.3     | 16.1    |
> | **Qwen2.5-VL-72B-Instruct**| 84.6      | 5.6   | 22.7    | 4.2     | 14.2    |
> | **CheXagent**             | 74.6      | 0.3   | 4.7     | 0.0     | 4.3     |
> | **MedVLM-R1**             | 82.1      | 5.9   | 16.8    | 2.7     | 12.1    |
> | **ChestX-Reasoner**       | 82.2      | 4.8   | 14.4    | 4.1     | 11.5    |
> | **Qwen2.5-VL-7B-Instruct**| 84.5      | 7.2   | 22.4    | 4.2     | 14.0    |
> | **GPT-4o**                | 86.2      | 11.1  | 30.4    | 6.4     | 18.4    |
> | **GPT-4o-mini**           | 85.2      | 6.7   | 22.8    | 3.8     | 14.2    |
> | **DeepSeek-VL2**          | 82.3      | 4.4   | 14.9    | 3.0     | 10.6    |
> | **Chiron-o1-2B**                    | 86.1      | 8.7   | 25.6    | 6.4     | 17.4    |
> | **Chiron-o1-8B**                    | 86.2      | 10.7  | 26.7    | 7.4     | 17.8    |
>
> | Models (Impression Generation)    | BERTScore | BLEU  | ROUGE-1 | ROUGE-2 | ROUGE-L |
> |-----------------------------------|:---------:|:-----:|:-------:|:-------:|:-------:|
> | **HuatuoGPT-Vision-7B**           | 83.5      | 2.8   | 10.6    | 1.8     | 7.5     |
> | **Qwen2.5-VL-72B-Instruct**       | 83.3      | 2.8   | 10.2    | 2.2     | 6.9     |
> | **CheXagent**                     | 81.4      | 7.3   | 9.8     | 0.0     | 9.8     |
> | **MedVLM-R1**                     | 81.5      | 1.9   | 6.1     | 0.5     | 4.8     |
> | **ChestX-Reasoner**               | 83.6      | 4.5   | 12.3    | 4.0     | 10.8    |
> | **Qwen2.5-VL-7B-Instruct**        | 83.6      | 3.7   | 10.0    | 2.0     | 7.2     |
> | **GPT-4o**                        | 84.5      | 4.1   | 13.9    | 3.0     | 9.8     |
> | **GPT-4o-mini**                   | 83.7      | 2.5   | 9.7     | 1.4     | 6.6     |
> | **DeepSeek-VL2**                  | 82.4      | 2.6   | 7.1     | 0.9     | 6.3     |
> | **Chiron-o1-2B**                  | 83.9      | 3.2   | 9.5     | 2.0     | 7.8     |
> | **Chiron-o1-8B**                  | 83.8      | 3.7   | 10.3    | 2.0     | 8.1     |
>
> **Re weakness 4:** Thank you for your constructive critique. We find the reviewer’s suggestion highly interesting and worthy of exploration. After validating multiple cases, experimental results consistently demonstrate that incorrect guidance or prompts are insufficient to lead Chiron-o1 to generate unreasonable reasoning processes. Here, we present an example with the question: “Teacher, based on the imaging findings of dilated small and large bowel with a transition point at the splenic flexure showing an ‘apple core’ lesion, what is the most likely diagnosis causing this patient’s large bowel obstruction? (Supplementary information is as follows: Gender: Male, Age: 40 years, Chief complaint: Abdominal bloating.)” The ground truth is “obstructing splenic flexure colon cancer.” Due to the constraints of the rebuttal response, we are unable to display the image here. If we intentionally provide an incorrect prompt in the question, such as “suggestive of bowel obstruction caused by colonic lymphoma,” Chiron-o1 can still filter out the erroneous information and arrive at the final diagnosis of “splenic flexure colon carcinoma,” which is consistent with the ground truth. Therefore, we have reason to believe that Chiron-o1 is capable of resisting interference from incorrect information. The detailed reasoning process and specifics will be included in the revised manuscript.
>
> **Re question 1:** Thank you for your attention. We will make the code, model, and data for reproducing MICS publicly available.
>
> **Re question 2:** We are grateful for your constructive feedback. Your perspective on this issue aligns closely with our own. Indeed, medicine is a highly rigorous discipline, and methods lacking medical logic are unacceptable, as they are irresponsible to human health. We would like to kindly clarify that the process of constructing CoT data with MICS is supervised and informed by real clinical data. Specifically, all case data from Radiopaedia are raw data verified by clinicians from medical centers across various countries and deemed valuable for learning. We collected information such as indications, age, gender, imaging interpretations, abnormal target localization, and case discussions to ensure the reliability of the MMRP dataset constructed with MICS. To further ensure the clinical logic of the data, we recruited three experienced radiology doctoral students to clean the MMRP data, retaining only high-quality CoT data. Additionally, we have noted related work on using knowledge graphs to generate CoT (e.g., MedReason). However, such work focuses solely on the text modality. In the future, we plan to adapt the paradigm of using knowledge graphs to construct CoT for multimodal medical reasoning.
>
> **Re question 3:** Thank you for highlighting this issue.  Please see our detailed answer to this question in response to weakness 2.

---

> ### Comment · Reviewer_kRLm · 2025-08-05
> **Official Comment by Reviewer**
>
> Thank you for your response. The additional discussion and experiments have largely addressed my main concerns, and I appreciate the authors’ efforts in this regard. I encourage the authors to further clarify these points in the final version. Additionally, with respect to weakness 2, using only a single example to verify whether the model relies on imaging features is not sufficiently robust. I recommend providing more comprehensive and rigorous validation in the final version.

---

> > ### Author Response · Authors · 2025-08-05
> >
> > Thank you for your efforts in the review process and for providing valuable feedback. Regarding weakness 2, we have actually conducted some similar qualitative experiments. However, due to the word limit in the rebuttal, we only presented one representative example. Ultimately, we will elaborate on your suggestions in the revised manuscript. Thank you again for your dedication to our work.

---

### Official Review · Reviewer_kkRL · 2025-07-23

**Clarity:** 3
**Significance:** 3
**Originality:** 4
**Rating:** 5
**Confidence:** 4

**Summary:**

This paper introduces **MICS**, a novel strategy for generating a high-quality CoT dataset from VQA samples. Using MICS, the authors curate **MMRP**, a medical reasoning dataset for SFT of language models. The authors also train **Chiron-o1**, a new multimodal medical reasoning model, through a three-stage training using MMRP and evaluate it on different datasets.

**Questions:**

1. Could the authors clarify the versions of the baseline models used for evaluation? Specifically, the paper mentions that Gemini 2.5 Pro is used as a mentor model in the MICS process, but in the baseline comparison (e.g., Table 1), the cited paper corresponds to Gemini 1.5 Pro. Why is a stronger model used for MCIS but not included as a baseline?

2. There appear to be discrepancies between the numbers reported in Table 1 and those in the original papers. For example, the LLaVA-Med paper reports accuracies of 61.52, 84.08, and 37.95 on VQA-RAD, SLAKE, and PathVQA (based on open evaluation for LLaVA-based version), whereas the numbers shown in Table 1 differ. Could the authors explain the source of these discrepancies? Were all baselines re-evaluated under a consistent setup?

3. For the VQA benchmarks, what evaluation protocol was used, open-ended or multiple-choice (closed-ended) answering? This distinction significantly affects accuracy metrics and should be clarified for fair comparison.

4. Have the authors explored the effect of reinforcement learning (RL) fine-tuning after the three-stage SFT process on Chiron-o1? Given the success of GRPO in Med-R1 and MedVLM-R1, it would be interesting to assess whether RL could further improve the reasoning capabilities of Chiron-o1.

5. There are several issues reported regarding a lack of clear visual understanding in multimodal language models, where they often fail to capture fine-grained image details, which is a crucial ability in the medical domain. Have the authors investigated whether the MICS strategy helps mitigate this issue? Does Chiron-o1 still suffer from these limitations?

6. It is not clear whether the intern models in the MICS process have access to the image input. Based on Figure 2, it appears they are only given the question and partial reasoning steps. If they do not see the image, could this harm their performance?

**Ethical Concerns:**

["NO or VERY MINOR ethics concerns only"]

**Final Justification:**

The discussion and additional experiments provided in the rebuttal add clarity to the paper and help strengthen the evaluation section. I believe the MICS method, MMRP dataset, and Chiron-o1 model represent valuable contributions to the field of medical multimodal reasoning. That said, the evaluation could still benefit from comparisons with more recent open-source models. Overall, despite some minor weaknesses, I find this to be a technically strong and well-executed paper.

**Limitations:**

Yes.

**Paper Formatting Concerns:**

There were no paper formatting concerns.

**Quality:**

3

**Strengths And Weaknesses:**

**Strengths**

- **MICS** is a novel and well-motivated framework for generating high-quality chain-of-thought (CoT) data. It provides a generalizable method that can be applied to existing datasets to enrich them with step-by-step reasoning traces.
- The authors introduce **MMRP**, a multimodal medical reasoning dataset, which could serve as a valuable resource for training and evaluating medical MLLMs.
- The paper includes thorough **ablation studies** that effectively demonstrate the impact of MICS and their proposed three-stage training strategy. These analyses help validate the importance of each design choice.

**Weaknesses**
- The paper lacks clarity in some aspects of the **evaluation setup** and **implementation details**. For example, it is unclear which exact versions of baseline models (e.g., **LLaVA-Med**, **Gemini**) were used in comparison. Additionally, important training details such as the number of GPUs used, total GPU hours, and compute infrastructure are missing (information that is required by the checklist).
- The **evaluation scope could be improved**. Since Chiron-o1 is trained on the datasets used in Table 1, it would be more convincing to include evaluations on **unseen datasets** such as *OmniMedVQA [1]* and *MediConfusion [2]*, to better assess generalization. Moreover, the absence of **proprietary models** (e.g., GPT-4o, Gemini 2.5 Pro) in Table 2 is not well justified, even though the table focuses on medical reasoning, including these models would provide useful reference points.
- While the proposed **MICS** strategy is promising, the paper would benefit from a more detailed analysis of **failure cases** or **limitations**. For example, in what types of reasoning tasks does MICS fail to generate effective paths? Are there modalities or body systems where it underperforms?

**References**
[1] Cheng, T., et al. *OmniMedVQA: A Benchmark for Generalized Medical Visual Question Answering*. CVPR, 2024.
[2] Sepehri, M. S., et al. *MediConfusion: Can you trust your AI radiologist? Probing the reliability of multimodal medical foundation models*. ICLR 2025.

---

> ### Author Rebuttal · Authors · 2025-07-31
>
> Thank you for your insightful review. We’re delighted that the reviewer finds MICS a “novel and well-motivated framework” for generating high-quality CoT data and appreciates MMRP as a “valuable resource” for medical MLLMs, supported by “thorough ablation studies.” Please see our response to the reviewer’s concerns below.
>
> **Re weakness 1:** Thank you for highlighting this issue. In Table 1, we used LLaVA-Med-v1.5-Mistral-7B and Gemini 1.5 Pro as baseline models. During training, we fine-tuned Chiron-o1 using eight A100 GPUs (accelerated with DeepSpeed ZeRO-1), with the three training stages taking 12 hours, 12 hours, and 48 hours, respectively. More detailed supplementary information will be included in the updated manuscript.
>
> **Re weakness 2.1:** Thank you for pointing this out. Although we have already achieved SOTA performance on the out-of-domain benchmark MMMU (H&M), we conducted further tests on additional unseen benchmarks (OmniMedVQA and MediConfusion) to validate the robust generalization of Chiron-o1.
>
> As shown in table below, Chiron-o1 demonstrates strong performance on the out-of-domain benchmark OmniMedVQA, confirming its robust VQA and reasoning capabilities.
> | Model                     | CT   | FP   | MRI  | OCT  | Der  | Mic  | X-Ray | US   | Avg. |
> |:-------------------------:|:----:|:----:|:----:|:----:|:----:|:----:|:-----:|:----:|:----:|
> | **Med-Flamingo**           | 34.6 | 33.3 | 27.5 | 26.0 | 28.3 | 28.1 | 30.1  | 33.2 | 30.2 |
> | **RadFM**                  | 33.3 | 35.0 | 22.0 | 31.3 | 36.3 | 28.0 | 31.5  | 26.1 | 30.5 |
> | **LLava-Med-7B**           | 25.3 | 48.4 | 35.9 | 42.1 | 45.2 | 44.0 | 31.7  | 83.7 | 44.5 |
> | **Qwen-VL-Chat**           | 51.5 | 45.4 | 43.9 | 54.0 | 55.4 | 49.5 | 63.1  | 33.5 | 49.5 |
> | **Yi-VL-34B**              | 39.8 | 57.2 | 51.4 | 70.5 | 54.5 | 61.4 | 40.5  | 40.5 | 54.9 |
> | **LLava-v1.6-7B**          | 40.1 | 39.5 | 54.8 | 58.4 | 54.0 | 48.8 | 53.3  | 47.9 | 49.6 |
> | **LLava-v1.6-13B**         | 40.0 | 43.6 | 47.4 | 63.2 | 58.0 | 50.5 | 59.6  | 42.6 | 50.6 |
> | **LLava-v1.6-34B**         | 50.6 | 63.4 | 60.9 | 68.4 | 65.7 | 62.8 | 74.7  | 44.5 | 61.4 |
> | **LLava-v1.5-LLaMA3-8B**   | 33.0 | 49.7 | 53.8 | 76.0 | 63.1 | 48.4 | 56.6  | 31.2 | 48.8 |
> | **HuatuoGPT-Vision-7B**    | 61.6 | 80.2 | 65.1 | 86.3 | 71.6 | 67.4 | 81.4  | 87.4 | 75.1 |
> | **HuatuoGPT-Vision-34B**   | 60.8 | 85.5 | 66.5 | 90.0 | 74.0 | 71.3 | 83.8  | 81.7 | 76.7 |
> | **Chiron-o1-2B**           | 72.0 | 81.0 | 85.5 | 84.5 | 68.5 | 73.0 | 75.5  | 57.0 | **74.6** |
> | **Chiron-o1-8B**           | 79.5 | 85.0 | 87.5 | 89.0 | 76.0 | 81.5 | 84.5  | 49.5 | **79.1** |
>
> Additionally, we conducted tests on the intriguing MediConfusion dataset, which uses pairs of similar images sharing the same question and options to assess whether the model can accurately distinguish medical images. The results in table below indicate that Chiron-o1 outperforms most general and medical models but still lags behind large-scale closed-source models. In the future, we aim to collect more high-quality annotated real-world medical imaging data to further enhance the model’s ability to discriminate medical images.
> | Model                          | Set accuracy (%) | Confusion (%) | Individual accuracy (%) |
> |:------------------------------:|:----------------:|:-------------:|:-----------------------:|
> | **RadFM**                      | 0.57             | 97.54         | 35.90                   |
> | **LLaVA-Med-v1.5 (Mistral 7B)**| 0.00             | 100.00        | 23.58                   |
> | **Gemini 1.5 Pro**             | 19.89            | 58.52         | 51.14                   |
> | **BLIP-2 (Opt 2.7B)**          | 0.57             | 92.19         | 22.16                   |
> | **GPT-4o**                     | 18.75            | 75.00         | 56.25                   |
> | **MedFlaming**                 | 1.14             | 98.75         | 47.73                   |
> | **Claude 3 Opus**              | 8.52             | 84.09         | 50.85                   |
> | **LLaVA-v1.6 (Mistral 7B)**    | 8.52             | 85.47         | 50.57                   |
> | **Chiron-o1-8B**               | 10.22            | 84.91         | 53.73                   |
>
> **Re weakness 2.2:** Thank you for raising this important point. Similarly, we have supplemented Table 2 in the manuscript by including experimental results for some proprietary models. The additional results show that Chiron-o1-8B can achieve performance comparable to large-scale closed-source models on reasoning-focused benchmarks. In the future, we aim to further refine MICS to collect higher-quality CoT data, thereby surpassing current performance levels.
>
>
>
> | Model       | Text  | MXQA-R | MXQA-U | R-ACC | R-Bert | R-MICS |
> |-----------|:-----:|:------:|:------:|:-----:|:------:|:------:|
> | **GPT-4o**        | 92.4  | 30.0   | 27.2   | 55.8  | 85.8   | 46.5   |
> | **Gemini 2.5 Pro**| 90.8  | 29.5   | 25.8   | 54.6  | 86.0   | 45.7   |
> | **Chiron-o1-8B**  | 92.1  | 23.3   | 25.1   | 58.4  | 90.4   | 49.4   |
>
> > **Abbreviations**:
> > - `Text`: MMRP (Pure Text)
> > - `MXQA-R`: MedXpertQA-MM (Reasoning)
> > - `MXQA-U`: MedXpertQA-MM (Understanding)
> > - `R-ACC`, `R-Bert`, `R-MICS`: MMRP (Reasoning) with ACC, Bert-Score, and MICS-Score respectively
>
> **Re weakness 3:** Thank you for your valuable perspective. Following your insightful suggestion, we conducted a thorough review of the constructed CoT data and identified several cases where reasoning failed. Through comprehensive analysis, we summarized several key issues: (a) When the resolution of the associated images is low, the model struggles to interpret them adequately, leading to the failure of MICS. (b) Since the collected data are real clinical cases uploaded by doctors from various hospitals on the Radiopaedia website, we found that a small number of cases contain discrepancies between the annotation information and the content displayed in the images (possibly due to clinicians’ errors). This also hinders MICS from effectively searching for high-quality CoT data. Furthermore, based on the evaluation results on OmniMedVQA from “Re weakness 2.1,” the scarcity of collected ultrasound modality data likely contributes to the model’s performance decline. Moving forward, we plan to collect more real-world clinical cases to expand the MMRP dataset, thereby addressing the aforementioned issues.
>
> **Re question 1:** Thank you for your critical and insightful feedback. We will include the versions of the baseline models in the updated manuscript, as mentioned in “Re weakness 1.” Additionally, due to our oversight, we did not initially include the more powerful closed-source model Gemini 2.5 Pro as a baseline for comparison. We have conducted further comparative experiments, with the results presented below:
>
> | Model             | vqa-rad | slake | pathvqa | pmcvqa | mmmu | Avg  |
> |:-----------------:|:-------:|:-----:|:-------:|:------:|:----:|:----:|
> | **Gemini 1.5 Pro**| 60.3    | 72.6  | 70.3    | 52.3   | 47.9 | 60.7 |
> | **Gemini 2.5 Pro**| 71.3    | 80.5  | 74.1    | 61.1   | 57.1 | 68.8 |
> | **Chiron-o1-8B**  | 76.8    | 83.2  | 74.0    | 57.5   | 54.6 | 69.2 |
>
> **Re question 2 and 3:** Thank you for your constructive critique. Regarding the experimental results in Table 1 of the manuscript, we would like to kindly clarify that we adopted the test set splitting standard from HuatuoGPT-Vision, which differs from that of LLaVA-Med. Ultimately, our experimental results are consistent with those reported in HuatuoGPT-Vision. To ensure fairness in the comparison, we adopted the evaluation settings of HuatuoGPT-Vision (whose code is open-source) when testing Chiron-o1, InternVL3, and other models. All experimental results in Table 1 were evaluated using a unified standard for close-ended questions.
>
> **Re question 4:** Thank you for your suggestion, which strengthens our work. As you noted, reinforcement learning (RL) methods such as GRPO, PPO, RLOO, and REINFORCE++ have already shown significant impact in the field of medical reasoning. Given that Chiron-o1 has developed robust reasoning capabilities through multi-stage training, we fully agree that using it as a baseline and applying RL algorithms could potentially unlock even more robust and powerful reasoning abilities. This is already part of our future work plan, and we look forward to sharing updates in next version of Chiron-o1.
>
> **Re question 5:** Thank you for your valuable perspective. We had indeed taken note of the issue you raised during the data collection process. Given that fine-grained visual understanding is critical for MLLMs to address various tasks, particularly reasoning tasks, we prioritized collecting data annotated by clinicians to include key diagnostic information (such as illustrated interpretations and case discussions) that align closely with the detailed features presented in the images. Through the construction of reasoning data using MICS and multi-stage training, we observed improvements in the model’s visual understanding, as evidenced by the experimental results and case studies described in the manuscript. However, we acknowledge that these limitations have not been fully overcome, a challenge also faced by general MLLMs that remains to be addressed. In the future, we plan to tackle this issue by expanding the collection of high-quality CoT data and refining our data construction and model training strategies.
>
> **Re question 6:** Thank you for your kind reminds. In fact, the intern model does have access to images. We will revise Figure 2 in the updated manuscript to avoid the misunderstanding.

---

> > ### Comment · Reviewer_kkRL · 2025-08-03
> >
> > Thank you for the detailed response and for sharing the results of the new experiments. The clarifications effectively address the ambiguities I had noted, and the additional evaluations strengthen the manuscript.
> >
> > I did notice that the updated OmniMedVQA evaluation covers only 8 out of the 12 modalities. Could you clarify the rationale behind this selection?

---

> > > ### Author Response · Authors · 2025-08-04
> > >
> > > Thank you for your thorough observation. To ensure a fair comparison of all experimental results in the manuscript, we consistently followed the test set splitting standard and evaluation settings of HuatuoGPT-Vision (eight modalities). Therefore, we evaluated only eight imaging modalities in this context.

---

> > > > ### Comment · Reviewer_kkRL · 2025-08-04
> > > >
> > > > Thank you for the clarification. The discussion and additional experiments provided in the rebuttal are satisfactory, and as a result, I am increasing my score to 5. Overall, I believe the MICS method, MMRP dataset, and Chiron-o1 model are valuable contributions to the field. That said, I still think the evaluation section could be further improved by including more recent open-source models such as LLaVA-OneVision, Qwen-VL 2.5, and others.

---

> > > > > ### Author Response · Authors · 2025-08-05
> > > > >
> > > > > Thank you for your valuable feedback and support! We appreciate your recognition of our work and will incorporate additional experiments with recent open-source models to strengthen the evaluation section in the revised version.

---

### Official Review · Reviewer_mcTK · 2025-07-23

**Clarity:** 3
**Significance:** 3
**Originality:** 2
**Rating:** 5
**Confidence:** 3

**Summary:**

This paper introduces Chiron-o1, a multimodal large language model for medical reasoning, along with a novel training methodology called Mentor-Intern Collaborative Search (MICS).
Key Contributions:
1. MICS Strategy: A collaborative framework where "mentor" models (GPT-4o, Gemini) generate reasoning steps while "intern" models (smaller open-source models) evaluate their quality. The system iteratively searches for effective reasoning paths by having interns complete reasoning from mentor-generated steps and selecting paths based on success rates.
2. MMRP Dataset: A three-part multimodal medical dataset containing:
57,630 filtered, text-only QA for basic knowledge
5,878  image-text alignment pairs
8,328 complex reasoning scenarios across clinical contexts
3. Chiron-o1 Model: Trained using 3-stage curriculum learning (text QA ->  image-text alignment -> complex reasoning with MICS-generated data), achieving good performance on medical VQA and reasoning benchmarks.
The core innovation is MICS, which creates high-quality medical reasoning data through collaborative model evaluation rather than single-model generation or expensive human annotation. This enables more rigorous reasoning paths that significantly improve performance on complex clinical scenarios compared to existing medical MLLMs.

**Questions:**

1. Computational Costs of MICS
Question: Can you clarify the computational resources, API costs, and environmental impact for training Chiron-o1 with MICS? How many API calls to GPT-4o/Gemini were used, and what's the approximate cost per reasoning path generated? Could you achieve similar results using only open-source models?

Evaluation criteria: Clearly describing computational costs and considering cheaper alternatives would improve the paper's practical value.

2. Medical Bias and Safety
Question: How do you ensure MICS doesn't increase biases or mistakes from the original medical data? Did you check if the mentor-intern method introduces systematic biases? Could you share details about dataset demographics, representation of diseases across different groups (e.g., malaria, sickle cell, emerging diseases), and methods for identifying or preventing bias? Did medical experts help you assess bias risks?

Evaluation criteria: Clearly discussing bias risks and showing concrete steps to reduce them would make the approach safer and more trustworthy.

3. Clinical Validation and Next Steps
Question: What are your plans for clinical validation? Did medical professionals help confirm the clinical usefulness of the reasoning patterns generated? Can you outline specific next steps required for clinical evaluation?

Evaluation criteria: Clearly outlining plans for clinical validation and showing involvement from medical professionals would strengthen confidence in real-world applicability.

4. Comparison with Human Medical Reasoning
Question: How does Chiron-o1’s reasoning quality compare to human medical experts? Can you highlight areas where the model performs particularly well and situations where human oversight is still necessary?
Evaluation criteria: Provide insights from subject matter experts to support comparisons and strengthen your conclusions.

**Ethical Concerns:**

["NO or VERY MINOR ethics concerns only"]

**Final Justification:**

Chiron-o1 combines curriculum learning with a mentor–intern framework for medical visual reasoning, achieving solid gains without the use of reinforcement learning (RL). The multi-mentor/intern approach introduces diversity in reasoning paths and has been thoroughly tested. In response to request for comparisons to other tree-based methods, evaluations using multiple models to reduce potential bias, and a clearer explanation of how intermediate reasoning steps are assessed, the authors provided additional comparisons, multi-model evaluations, and a detailed explanation of how their MICS framework maintains reasoning quality. Overall, the work is quite readable, introduces new approaches, and should be accepted.

**Limitations:**

Technical Limitations
Could discuss computational costs and API expenses required, which could limit accessibility
Heavy reliance on closed-source models (GPT-4o, Gemini) creates vulnerability to API changes, pricing modifications, or service discontinuation.
Limited discussion of whether BERT-Score and MICS-Score truly capture medical reasoning quality, or potential biases in the evaluation framework. Bringing in expert opinion could help as well.
Generalization boundaries: Insufficient analysis of when/where the approach might fail or perform poorly (ie rare diseases (though examples quite rare), emerging diseases, different medical systems).
Missing Societal Impact Discussion
Could discuss accountability when AI-generated reasoning leads to incorrect medical decisions or diagnoses.
Environmental impact: The computational intensity of MICS, requiring multiple API calls to large models and extensive inference, contributes to climate and environmental risks.
Bias amplification and error reproduction: The training data from existing sources may perpetuate existing medical biases, diagnostic errors, and systematic mistakes already present in medicine.
Healthcare worker deskilling and automation bias: Risk that medical professionals may become overly dependent on AI reasoning, potentially diminishing critical thinking skills and leading to automation bias where clinicians uncritically accept AI-generated conclusions.
Would help to discuss regulatory hurdles expected for clinical deployment

Missing Safeguards Discussion
Clinical validation: Need further clinical evaluation before deployment, could discuss
Human oversight mechanisms: How to ensure appropriate human supervision
Bias monitoring: Strategies for detecting and mitigating biased reasoning patterns
Failure mode detection: Methods for identifying when the system is operating outside its capacity

**Paper Formatting Concerns:**

No all looks good.

**Quality:**

3

**Strengths And Weaknesses:**

1. Strengths

a. Quality
Design: The paper conducts comprehensive evaluations across 7 benchmarks, including both in-domain (MMRP) and out-of-domain (MedXpertQA_MM) scenarios, demonstrating good generalization capabilities.
Ablation studies: Systematic analysis of MICS components, training strategies, and dataset configurations supports design.
Methodology: The framework here addresses a real problem in medical CoT data generation - the lack of quality evaluation mechanisms.
Consistent improvements: Chiron-o1 achieves gains across multiple benchmarks compared to baseline models.

b. Clarity
Well-structured presentation: The paper flows well.
Clear technical exposition: The MICS algorithm is explained well and visualizations help provide more clarity.

c. Significance
Addresses healthcare need: Medical reasoning is a complicated process, not always captured in many models; improved AI capabilities could have substantial impact.
Scalable approach: MICS provides a framework for generating high-quality reasoning data that could be applied beyond medical domains.
Clear Results: Demonstrates clear advancement over existing medical MLLMs, particularly in reasoning tasks.

d. Originality
New collaborative search approach: The mentor-intern framework for reasoning path evaluation is creative and well-motivated by educational analogies.
Evaluation metric: MICS-Score provides a set way to assess reasoning path quality using multiple model perspectives.
Comprehensive dataset construction: The three-stage MMRP dataset design with medical content could help with other models

2. Weaknesses

a. Quality
The MMRP dataset would benefit from clinician sampling and testing. There is the risk of data not being as representative or as clinically logical as expected.
Limited theoretical analysis: The paper lacks theoretical justification for why the mentor-intern collaboration should work better than alternatives
Computational cost concerns: MICS requires multiple API calls to expensive models (GPT-4o, Gemini) and numerous intern model evaluations, making it potentially too costly in multiple respects for large-scale deployment. The expense should be recognized to include environmental cost, as these models risk using high amounts of electricity, water and  furthering climate change and the benefit may not be as clear when paired with downstream effects. Showing what the relative costs and benefits may be, even if very rough, would help.
Evaluation limitations:
Test size for benchmarks
BERT-Score for semantic similarity may not capture medical reasoning quality adequately

b. Clarity
Complex methodology: While well-explained, the MICS algorithm involves many components that may be difficult to reproduce reliably.

c. Significance
Incremental: While effective, the core contribution builds incrementally on existing CoT and collaborative reasoning approaches rather than being an entirely new approach.
Comparison limitations: Could compare with more recent medical reasoning frameworks

d. Originality
Limited novelty of each piece: Novelty ia from the combination of study parts
Dataset overlap concerns: The MMRP dataset is built on existing medical case data, often used in such studies

---

> ### Author Rebuttal · Authors · 2025-07-31
>
> Thank you for your insightful review. We’re delighted that the reviewer finds the MICS strategy “creative and well-motivated,” praises the “comprehensive” evaluations and “clear” presentation. We have summarized and organized your comments and suggestions, and we provide detailed responses below.
>
> **Re bias and safety of MMRP (weakness 1 + question 2 + limitation 6):**  We appreciate your insightful comments. While our MMRP dataset, sourced from real Radiopaedia clinical cases, is valuable, we acknowledge the potential for errors or biases from manual annotation. To address this, we implemented initial data preprocessing (e.g., removing unclear cases/low-resolution images). Crucially, to mitigate systematic biases introduced by MICS, we recruited three experienced radiologists to meticulously clean the constructed CoT data, removing unreasonable or clinically invaluable entries. The table below presents statistical data for the MMRP dataset, including representative diseases like neurological tumors and infectious diseases.
> | Age Range   | Count |
> |:-----------:|:-----:|
> | 0–9         | 241   |
> | 10–19       | 262   |
> | 20–29       | 358   |
> | 30–39       | 410   |
> | 40–49       | 395   |
> | 50–59       | 374   |
> | 60–69       | 346   |
> | 70–79       | 237   |
> | 80–89       | 108   |
> | Unknown     | 32
>
> | Gender   | Count |
> |:--------:|:-----:|
> | Male     | 1,467 |
> | Female   | 1,301 |
> | Unknown  | 8     ||
>
> **Re theoretical analysis of MICS:** Thank you for this point. Unlike mainstream direct distillation, our MICS method constructs high-quality CoT data through collaborative interaction between mentor and intern models. Specifically, the more knowledgeable mentor model provides prompts (partial reasoning paths) that guide the weaker intern model to complete reasoning and achieve the correct answer, thereby demonstrating the value of the mentor's reasoning prefix—much like a teacher guiding a student. As Figure 4(a) of the manuscript illustrates, our MICS method is indeed more effective than other CoT annotation approaches.
>
> **Re computational cost of MICS (weakness 3 + question 1 + limitation 1 + limitation 5):** We appreciate your feedback. For Intern models, inference is handled by local deployment, with a single H100 GPU (80G) sufficient for eight models. Mentor model API costs, which are the primary concern, depend on the number of mentor models and search depth; as shown in Table 1 (API cost) and Table 2 (API calls), API consumption increases with more mentor models based on our analysis of 300 sampled cases. We acknowledge the significant cost-performance gap between closed-source and open-source models, with our CoT data currently relying on the superior reasoning of closed-source mentors (e.g., GPT, Gemini). Consequently, we advocate balancing MICS performance with resource usage and hope future open-sourcing of powerful models will enhance strategies like MICS. (We set mentor_1, mentor_2, mentor_3, and mentor_4 as ChatGPT-4o, Gemini 2.5 Pro Preview, Qwen2.5-VL-72B-Instruct, and Qwen2-VL-72B-Instruct, respectively.)
>
> | MICS (cost/$)     | ChatGPT-4o | Gemini 2.5 Pro Preview | Qwen2.5-VL-72B-Instruct | Qwen2-VL-72B-Instruct | Deepseek-V3 (evaluation) |
> |-------------------|------------|------------------------|-------------------------|-----------------------|--------------------------|
> | **1 mentor**        | 3.67       | -                      | -                       | -                     | 0.07                     |
> | **2 mentors**       | 4.76       | 4.11                   | -                       | -                     | 0.24                     |
> | **3 mentors**       | 5.26       | 4.78                   | 0.39                    | -                     | 0.52                     |
> | **4 mentors**       | 5.76       | 5.12                   | 0.43                    | 0.71                  | 1.07                     |
>
> | MICS (call_numbers) | ChatGPT-4o | Gemini 2.5 Pro Preview | Qwen2.5-VL-72B-Instruct | Qwen2-VL-72B-Instruct | Deepseek-V3 (evaluation) |
> |---------------------|------------|------------------------|-------------------------|-----------------------|--------------------------|
> | **1 mentor**          | 317        | -                      | -                       | -                     | 1,159                    |
> | **2 mentors**         | 409        | 381                    | -                       | -                     | 3,903                    |
> | **3 mentors**         | 452        | 443                    | 428                     | -                     | 8,568                    |
> | **4 mentors**         | 504        | 490                    | 487                     | 490                   | 16,753                   |
>
> **Re evaluation limitations (weakness 4 + limitation 2):** Thank you for your valuable and detailed comments. We fully agree that Bert-Score cannot fully reflect the quality of medical reasoning. Therefore, we proposed MICS-Score to address the shortcomings of existing medical reasoning evaluation metrics. Unlike other metrics, MICS-Score does not focus on character matching or semantic similarity but evaluates the mentor’s reasoning prefixes through the intern model’s performance. The advantage of this metric lies in its effectiveness in quality control (a low-quality reasoning path prompt is unlikely to guide the intern model to correct reasoning) and the robustness of the evaluation process (using multiple intern models to stabilize evaluations and avoid extreme cases). In the future, we plan to incorporate more real expert opinions as part of the evaluation metrics.
>
> **Re reproduce of MICS:** Thank you for your insightful comment. We will publicly release the code and model weights for reproducing MICS to ensure that anyone can replicate the entire process of constructing CoT data.
>
> **Re limited novel of MICS:** Thank you for your constructive critique. We would like to kindly clarify that there is currently no method using collaborative reasoning to construct high-quality CoT data, and the proposal of MICS was inspired by tree search. We aim to transfer the concept of tree search to the domain of data construction, introducing a novel approach for high-quality CoT annotation. Specifically, partial reasoning paths output by the mentor model are evaluated by the intern model to filter out superior reasoning steps, and collaborative reasoning continues along these high-quality prefixes to generate complete reasoning processes. As shown in the manuscript results, our proposed MICS effectively enhances model reasoning performance and has been recognized as a novel data construction method by Reviewers eXAH and kkRL.
>
> **Re comparison limitations:** We are thankful for your helpful suggestion. We have recently noted MLLMs related to medical reasoning. To ensure fair comparisons, we evaluated ChestX-Reasoner on five common benchmarks using the same evaluation settings. The specific results, shown in the table below, clearly demonstrate that Chiron-o1 maintains a significant advantage over the latest medical reasoning models.
>
> | Model               | vqa-rad | slake | pathvqa | pmcvqa | mmmu | Avg  |
> |-------------------|:-------:|:-----:|:-------:|:------:|:----:|:----:|
> | **ChestX-Reasoner** | 70.9    | 70.0  | 66.7    | 38.5   | 49.5 | 59.1 |
> | **Chiron-o1-2B**    | 75.4    | 85.3  | 70.3    | 54.3   | 42.1 | 65.5 |
> | **Chiron-o1-8B**    | 76.8    | 83.2  | 74.0    | 57.5   | 54.6 | 69.2 |
>
> **Re dataset overlap:** Thank you for highlighting this issue. We fully understand your concern about potential overlap between the MMRP dataset and commonly used training data or benchmarks. Unlike datasets such as MedXpertQA, PMC-VQA, and OmniMedVQA, the MMRP dataset is sourced from the Radiopaedia medical teaching website, which contains real cases uploaded by doctors from medical institutions across various countries, offering high clinical value. To date, this data has not been used as publicly available training data or benchmarks. Therefore, we confirm that the constructed MMRP dataset does not have data leakage issues.
>
> **Re clinical validation (question 3 + limitation 8):** We acknowledge your points on clinical validation, deployment, and safety. Formal clinical validation will proceed in two phases: **Systematic Retrospective Validation:** Blinded evaluation by radiologists on an independent dataset, focusing on accuracy, logical coherence, clinical utility, and safety. **Simulated Clinical Reader Studies:** Comparing doctor performance with/without Chiron-o1 to quantify improvements in accuracy, confidence, and efficiency. We will actively pursue these plans in future work.
>
> **Re generalization boundaries (question 4 + limitation 3):** Thank you for your valuable perspective. Through comprehensive qualitative analysis of existing case data, we have observed that Chiron-o1 does exhibit suboptimal generalization in certain special cases. For example, compared to human experts, the model tends to generate hallucinations when processing low-resolution images due to its inability to capture fine visual features. Additionally, as the MMRP dataset includes some rare diseases, Chiron-o1 finds it challenging to handle these cases without additional clinical information, requiring physician supervision. Honestly, we plan to address these issues in the future by further expanding the MMRP dataset to enhance the model’s reasoning performance.
>
> **Re societal impact (limitation 4 + limitation 7):** We concur on the importance of societal impact, accountability, bias, and regulation. Chiron-o1 is designed as a clinical decision support tool with verifiable reasoning to ensure human responsibility and prevent automation bias. Its interpretability also serves as an educational tool. This transparency, combined with rigorous phased validation, aims to fulfill future regulatory demands.

---

> > ### Comment · Reviewer_mcTK · 2025-08-06
> >
> > Thank you for the comprehensive rebuttal. The responses substantially address the core methodological questions around MICS collaborative reasoning and dataset construction quality. The detailed computational cost breakdown and additional comparative evaluations (ChestX-Reasoner, OmniMedVQA) strengthen the experimental foundation

---

> > > ### Author Response · Authors · 2025-08-06
> > >
> > > Thank you for your support of our work! We will elaborate on your suggestions in the revised manuscript to improve our paper. Thank you again for your valuable feedback.

---

### Official Review · Reviewer_eXAH · 2025-07-23

**Clarity:** 3
**Significance:** 3
**Originality:** 3
**Rating:** 4
**Confidence:** 4

**Summary:**

Chiron-o1 is a novel approach for training VLMs to be better at medical visual and reasoning tasks. Through the use of curriculum learning, the authors first train the model to become grounded with clinical cases and their corresponding treatments (stage I). In stage II, the model is then trained with images with their corresponding diagnoses. Finally, the model is trained with their novel mentor-intern collaborative search strategy where a more powerful mentor model is generates a reasoning prefix before the an intern model completes the remaining reasoning. If the intern model reasons well and is able to complete the chain frequently to arrive at the answer, then it is clear that the approach yields a successful thought.

**Questions:**

1. It's still not entirely clear why you had to incorporate VQA as a separate item for fine-tuning in Equation 6. I understand that the MMRP is  not enough which is why you added VQA, but you need to further explain this point, especially since MMRP includes stage 2 which involves fine-tuning with text-image pairs.

2. What is the performance of different number of intern/mentor models? Is it the case that a single intern/mentor model dominated the completion in most cases?

3. Can you report how does the number of API calls/tokens change compared to rejection fine-tuning with just the intern models?

**Ethical Concerns:**

["NO or VERY MINOR ethics concerns only"]

**Final Justification:**

I will give it a 4. The authors did a good job with maintaining coming up with a novel strategy for training VLM for medical reasoning without excessive dependency on RL and I think people should look at this approach to expand on it for other reasoning cases.

**Limitations:**

Yes.

**Quality:**

3

**Strengths And Weaknesses:**

Strengths:

1. Without using any reinforcement learning, the authors are able to get significant improvement with just fine-tuning data.

2. The mentor-intern collaboration is a novel approach which allows for diversity in thoughts and thinking by having multiple mentor and intern models attempt to complete the question.

3. Approach has been thoroughly benchmarked with general medical VQA and VQA requiring extensive reasoning, which demonstrates that the approach has a lot of merit in both simple and complex scenarios.

Weaknesses:

1. While the authors demonstrate that their model is better than other medical reasoning approaches, comparison against other tree-branching approaches would better help resolve the novelty of this approach.

2. DeepSeek-V3 is the only model being used for validation for the various reasoning chains, but other models should be used too to prevent biases.

3. Intermediate reasoning steps are not independently verified, they are only assessed based on if the final answer is correct or not.

---

> ### Author Rebuttal · Authors · 2025-07-29
>
> Thank you for your valuable feedback. We’re glad the reviewer finds Chiron-o1’s curriculum learning and mentor-intern strategy “novel” and appreciates its thorough benchmarking, showing “merit in both simple and complex scenarios.” Please see our response to the reviewer’s concerns below.
>
> **Re weakness 1:** Thank you for raising this insightful question. Our proposed MICS framework leverages the collaborative interaction between the mentor model and the intern model to construct high-quality multimodal Chain-of-Thought (CoT) medical reasoning data. In essence, it employs a tree search strategy, distinct from BFS or DFS, by using the MICS-Score as an evaluation metric to guide the expansion of the CoT data construction process. To further validate the effectiveness of MICS, we compared it with other tree-branch-related CoT construction method, such as Mulberry [1]. We randomly sampled 500 instances and constructed CoT data using both MICS and Mulberry. The quality of the resulting CoT data was then evaluated using the MICS-Score. The results, as shown in the table, demonstrate that MICS outperforms other methods in generating high-quality CoT data.
>
> | Methods   | MICS-Score |
> |:---------:|:----------:|
> | Mulberry  | 66.7       |
> | MICS      | 83.2         |
> 1. Yao, Huanjin, et al. "Mulberry: Empowering mllm with o1-like reasoning and reflection via collective monte carlo tree search." arXiv preprint arXiv:2412.18319 (2024).
>
> **Re weakness 2:** Thank you for raising this point. We would like to kindly clarify that DeepSeek-V3 is used to compare the outputs of the intern model with the ground truth to evaluate the quality of the reasoning paths. We fully agree with your suggestion to use multiple models to avoid bias. We randomly sampled 500 instances and evaluated them using different evaluation models. As shown in the table below, the variance in MICS-Score calculated across different models is relatively small, indicating no significant bias.
>
> | Models           | MICS  | Mulberry |
> |:----------------:|:-----:|:--------:|
> | GPT-4o           | 83.7  | 67.1     |
> | Gemini 2.5 Pro   | 83.1  | 66.8     |
> | DeepSeek-V3      | 82.7  | 66.4     |
> | **(variance)**   | 0.253 | 0.123    |
>
> **Re weakness 3:** Great question. We would like to elaborate on the evaluation of intermediate reasoning steps from two perspectives: **(a) CoT Construction:** The real-world clinical cases we currently collected include only the question, ground truth, and key frames of relevant medical imaging. The absence of intermediate reasoning processes makes direct supervision of the generated reasoning steps infeasible, a common challenge in the CoT field. Most methods rely on distilling closed-source models to extract reasoning processes for constructing CoT data. Building on this, we propose MICS, which searches for high-quality CoT data through a collaborative “mentor generation” and “intern validation” approach to ensure effective supervision of intermediate reasoning steps. **(b) Quality Evaluation:** The MICS-Score serves as a universal metric for assessing the quality of reasoning paths. Specifically, an effective and logically clear reasoning path should guide a smaller model (intern) to complete subsequent reasoning and arrive at the correct answer. Thus, we indirectly validate intermediate reasoning steps by verifying whether the intern model can be inspired by the existing reasoning process, addressing the challenge of evaluating reasoning processes.
>
> **Re question 1:** Thank you for your insightful comment. The Part 2 of the MMRP dataset consists of real clinical imaging interpretation data we collected, specifically in the form of image-caption pairs. Our goal was to incorporate more image-text alignment data during the training of Chiron-o1 to enhance the model’s imaging interpretation capabilities which are different from VQA capabilities. In fact, such imaging data directly interpreted by clinicians is relatively scarce. Therefore, we also gathered visual question-answering (VQA) data from various sources to improve the model’s performance and lay a foundation for enhancing its multimodal reasoning capabilities in the future.
>
> **Re question 2.1:** We appreciate your valuable suggestion. We randomly sampled 300 data instances and conducted comparative experiments by varying the number of mentor models (1 to 4) and intern models (2, 4, 6, 8) to assess their impact. The quality of the constructed CoT data under different settings was evaluated using the MICS-Score. From the experimental results shown in the table below, it is evident that the number of mentor models significantly influences the quality of the generated reasoning paths. A larger number of mentor models enhances the diversity of the MICS search process, thereby improving the effectiveness of the reasoning data. To balance API call costs, computational resources, and performance improvements, we adopted a standard MICS configuration with three mentor models and six intern models.
>
> |          | 1 mentor | 2 mentors | 3 mentors | 4 mentors |
> |:--------:|:--------:|:---------:|:----------:|:-----------:|
> | **2 interns**   | 85.8     | 88.0      | 90.0       | 91.0        |
> | **4 interns**  | 86.0     | 88.9      | 90.5       | 90.7        |
> | **6 interns** | 85.7     | 88.5      | 91.1       | 91.3        |
> | **8 interns**| 86.4     | 89.0      | 91.0       | 91.5        |
>
> **Re question 2.2:** Thank you for your thoughtful observation. We also noted this concern during the experiments and took measures to ensure that the constructed CoT data did not predominantly originate from a single mentor model. Therefore, we tracked the participation frequency of each mentor model during the construction of the MMRP Part 3 dataset. As shown in the table, the invocation counts across different mentor models are balanced, with no significant bias toward any single model. Additionally, we would like to kindly clarify that all intern models are fully involved in evaluating the generated reasoning paths, ensuring that there is no scenario where only a subset of intern models participates in the evaluation while others do not.
>
> |                      | ChatGPT-4o | Gemini 2.5 Pro Preview | Qwen2.5-VL-72B-Instruct |
> |:--------------------:|:----------:|:----------------------:|:-----------------------:|
> | Proportion (%)   | 34.4       | 33.4                   | 32.2                    |
>
> **Re question 3:** Thank you for pointing this out. During the construction of CoT data with MICS, all intern models (open-source) were deployed locally, and only the mentor models required API calls. Taking the experiments in “Re question 2.1” as an example, we set mentor_1, mentor_2, mentor_3, and mentor_4 as ChatGPT-4o, Gemini 2.5 Pro Preview, Qwen2.5-VL-72B-Instruct, and Qwen2-VL-72B-Instruct, respectively. The API call counts and associated costs are presented in the table below. In summary, while a greater number of mentor models enhances the quality of the generated CoT data, it inevitably increases resource consumption and costs. We aim to find an optimal balance to maximize the effectiveness of MICS.
>
> |                     | 1 mentor | 2 mentors | 3 mentors | 4 mentors |
> |:-------------------:|:--------:|:---------:|:----------:|:-----------:|
> | API_call        | 317      | 790       | 1,304      | 1,973       |
> | API_cost ($)    | 3.6      | 6.4       | 7.5        | 9.0         |

---

> > ### Comment · Reviewer_eXAH · 2025-08-01
> >
> > I thank the authors for their responses and hope they will include these results in the final version. I will keep my score the same.

---

> > > ### Author Response · Authors · 2025-08-04
> > >
> > > We appreciate the reviewer’s comments on our paper, we will do our best to incorporate all the feedback and further strengthen our work.

---

### Decision · Program_Chairs · 2025-09-17

**Decision:**

Accept (poster)

**Comment:**

The authors present a study on advancing reasoning capabilities in multimodal large language models (MLLMs) for the medical domain. They suggest that lack of systematic approaches for constructing chain-of-thought (CoT) data as a key barrier to progress. To this end, they introduce Mentor-Intern Collaborative Search (MICS), a framework for generating and evaluating reasoning paths. MICS employs mentor models to guide initial reasoning steps, followed by intern models that extend these paths, with final selection based on an MICS-Score designed to assess reasoning quality. Building on this framework, the authors develop MMRP, a multi-task medical reasoning dataset with graded difficulty, and Chiron-o1, a medical MLLM trained via curriculum learning. Experimental results suggest that Chiron-o1, trained with data generated through MICS, outperforms prior models on multiple medical visual question answering and reasoning benchmarks. The work emphasizes contributions at both the methodological and resource levels, with datasets, models, and code promised for public release.

Reviewers consistently highlight several strengths: the originality of the mentor–intern paradigm, comprehensive benchmarking across multiple datasets, ablation studies validating design choices, and strong empirical performance gains without resorting to reinforcement learning. The paper is also praised for its clarity and well-structured presentation, with evaluations demonstrating both in-domain and out-of-domain generalization. Weaknesses include limited evaluation scope (initially lacking comparisons to some recent baselines), high computational costs of MICS, and concerns about the need for more rigorous clinical validation and expert involvement. Additionally, some reviewers noted incremental rather than radical novelty, missing details on evaluation protocols, and underexplored failure cases. Importantly, the authors’ rebuttal successfully addressed most concerns: they added comparisons to alternative tree-search methods and recent baselines, provided detailed computational cost analyses, demonstrated multi-model evaluations to reduce bias, and clarified intermediate reasoning assessment. They also outlined plans for clinical validation and safeguards for safe deployment. Overall, while the paper has some limitations in evaluation breadth and resource efficiency, is a nice addition to Neurips and I recommend acceptance.